# From Collapse to Control: Understanding and Extending Context Length in Emerging Hybrid Models via Universal Position Interpolation

**Haochen Shen**[1], **Davis Wertheimer**[2], **Zheng Wang**[1], **Garrett Goon**[2], **Derrick Liu**[2],
**Naigang Wang**[2], **Mudhakar Srivatsa**[2], **Raghu K. Ganti**[2], **Minjia Zhang**[1]
[1]SSAIL Lab, University of Illinois Urbana-Champaign [2]IBM Research, USA
{hshen14,zhengw10,minjiaz}@illinois.edu
{davis.wertheimer,goon,dliu}@ibm.com
{nwang,msrivats,rganti}@us.ibm.com

## Abstract

Hybrid Mamba-Transformer models have emerged as promising alternatives to Transformers, offering efficiency and competitive performance. However, they struggle to generalize beyond their training context windows, collapsing on long-context tasks. We provide the first systematic analysis of this failure, showing that it arises from uncontrolled state growth and uneven receptive field contributions across the hybrid architecture. Guided by this understanding, we introduce Universal Position Interpolation (UPI), a lightweight, training-free scaling method that unifies Mamba's cumulative decay with Transformer rotary frequency scaling. UPI selectively stabilizes unstable Mamba dynamics while rescaling Transformer encodings, controlling state growth and enabling reliable long-context generalization, with only a few auxiliary forward passes. Evaluation shows that UPI extends multiple state-of-the-art hybrid and pure Mamba models from 4K to up to 64K tokens on PG-19 perplexity, LongBench and RULER benchmarks, without sacrificing short-context accuracy. These findings establish the first principled bridge between context length extension on Transformers and state-space models and open a new direction for training-free context extension methods for emerging hybrid models.

## 1 Introduction

Large language models (LLMs) increasingly rely on long-context reasoning for applications such as document understanding (Liu et al., 2024a; Hsieh et al., 2024b; Bai et al., 2024), retrieval (Mohtashami & Jaggi, 2023; Kamradt, 2023; Hsieh et al., 2024a), and multi-turn dialogue (Li et al., 2017; Maharana et al., 2024). However, scaling to long contexts remains a fundamental challenge. Canonical transformer models capture long-range dependencies through attention, but incur quadratic computation and memory complexity, limiting their scalability on very long sequence lengths.

Hybrid Mamba-Transformer models have gained ground as a promising alternative (Lenz et al., 2025; Ren et al., 2024; Zuo et al., 2025). Interleaved transformer and selective state-space models (SSMs) (Gu & Dao, 2023; Dao & Gu, 2024) achieve strong performance at a fraction of computational cost. Models like Nemotron-H (NVIDIA et al., 2025b), Jet-Nemotron (Gu et al., 2025), Nemotron-Nano-2 (NVIDIA et al., 2025a), and Bamba (Chu et al., 2024; Ganti et al.) demonstrate strong performance against public Transformer LLMs (e.g., LLaMA-3.1) at substantially lower inference cost, making hybrid models a compelling design choice for next-generation long-sequence modeling.

While hybrid models deliver promising performance, their ability to generalize beyond the training context length remains largely unexplored. Our preliminary results suggest that hybrid models collapse when inputs exceed the training window. Bamba-9B-v2 (Ganti et al.), for example, is a state-of-the-art hybrid model with comparable performance to Llama-3.1-8B alongside 2-2.5x faster speed Yet as shown in Fig. 1, perplexity spikes when context length surpasses the 4K-token training length. Making hybrid models practical therefore requires solving the context length extension (CLE) problem: enabling models trained on shorter windows to generalize to longer contexts.

CLE methods exist for Transformers (e.g., PI (Chen et al., 2023), YaRN (Peng et al., 2024), LongRoPE (Ding et al., 2024; Shang et al., 2025)) and for Mamba (e.g., DeciMamba (Ben-Kish et al., 2025), StuffedMamba (Chen et al., 2025), MambaExtend (Azizi et al., 2025)), but to our knowledge none have been tested on hybrid models. LongMamba (Ye et al., 2025), the sole exception, only tests on a 1.2B scale model and provides no analysis into its failure. Thus, it remains unclear why hybrid CLE fails, or how to solve it. A second challenge lies in scalability. Many CLE methods rely on fine-grained token-wise control (Ben-Kish et al., 2025; Ye et al., 2025) or hours-long, model-specific, search-based rescaling of positional embeddings (Chen et al., 2023; LocalLLaMA, 2023a;b; Ding et al., 2024; Shang et al., 2025), and often require fine-tuning on long sequences. This

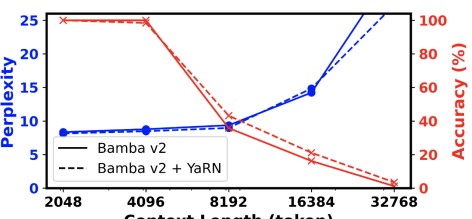

Figure 1: Performance of state-of-the-art hybrid model on long-context tasks. Blue lines denote perplexity($\downarrow$) on PG-19 validation set; red lines show retrieval accuracy($\uparrow$) on Needle-in-a-Haystack Single-Key-2 task from RULER (Hsieh et al., 2024a). Model: Bamba v2 (Ganti et al.), with 4k training context. YaRN applied without tuning.

is particularly costly, involving massively inflated activations, memory overhead and time. Meanwhile, training-free CLE remains largely unexplored for hybrid models.

Motivated by these challenges, we introduce *Universal Position Interpolation* (UPI), a closed-form CLE technique for scaling hybrid Mamba-Transformer models without retraining or inference overhead. We conduct the first systematic analysis of hybrid model scaling behavior, identifying uneven receptive field contributions and uncontrolled state growth as key factors behind long-context failure. Via selective rescaling, UPI prevents state explosion and restores balanced contributions. Most importantly, UPI requires no additional training, hyperparameter tuning, or modification of highly-fused kernels. By unifying Transformer position encoding with Mamba gate dynamics, UPI establishes the first principled, efficient, training-free CLE method tailored to hybrid models.

We evaluate UPI on PG-19, LongBench, and RULER benchmarks using state-of-the-art Mamba and hybrid models such as Mamba2, Bamba and Nemotron-H. UPI extends usable context length from 4K up to 64K tokens, delivering substantially lower perplexity and higher accuracy. Relative to architecture-specific CLE baselines, UPI achieves comparable or superior performance without training. Moreover, UPI requires substantially less effort to identify unstable heads than prior methods (e.g. LongMamba) and adds no inference-time overhead or model modifications.

## 2 BACKGROUND AND RELATED WORK

**The Emergence of Hybrid Mamba-Transformer Models.** Improving efficiency and scalability has been a central focus of LLM research. Beyond optimizing Transformer attention with approximations (Wang et al., 2020; Kitaev et al., 2020; Yuan et al., 2025) or high-performance kernels (Dao et al., 2022; Dao, 2024), alternative architectures such as RWKV (Bo, 2021; Sun et al., 2023), and S4 (Fu et al., 2023; Gu et al., 2022) achieve linear-time sequence modeling with reduced memory footprint and compute cost, but often struggle to match Transformers in accuracy when scaled to large tasks. More recently, Mamba (Gu & Dao, 2023; Dao & Gu, 2024) introduces a GPU-friendly linear recurrence that enables efficient modeling of long sequences while offering competitive accuracy (Wang et al., 2024; Shams et al., 2024; Zhu et al., 2024; Liu et al., 2024b; Schiff et al., 2024; Pióro et al., 2024). At the $t^{\text{th}}$ token $\mathbf{x}_t \in \mathbb{R}^d$, Mamba's state update recurrence can be simplified as: $\mathbf{h}_t = \bar{a}_t \mathbf{h}_{t-1} + \overline{\mathbf{B}}_t \mathbf{x}_t$, $\mathbf{y}_t = \mathbf{C}_t \mathbf{h}_t$. Forget gates $\bar{a}_t := \exp(-a\Delta_t) \in (0, 1)$ and input gates $\overline{\mathbf{B}}_t := \Delta_t \mathbf{B}_t \in \mathbb{R}_+^{h \times d}$ are discretized using Zero-Order Hold with a token-wise step size of $\Delta_t \in \mathbb{R}_+$. $\Delta_t$ governs update magnitude, by modulating the input and forget gates. Output gate $\mathbf{C}_t \in \mathbb{R}^h$ projects the updated hidden state to the output $\mathbf{y}_t \in \mathbb{R}^d$. This linear recurrence is computed via parallel scan (e.g., Hillis & Steele (1986)) on modern GPUs, requiring only $O(N)$ compute and $O(\log N)$ parallel depth over sequence length $N$. More detail can be found in Appendix B.

Although Mamba offer both a structured approximation to attention and highly parallel recurrence, pure Mamba models still lag behind Transformers in accuracy across a range of tasks. This performance gap has motivated hybrid architectures, such as Jamba (Lenz et al., 2025; 2024), Nemotron-H (NVIDIA et al., 2025b), Bamba (Chu et al., 2024), Jet-Nemotron (Gu et al., 2025), with compu-

tationally efficient Mamba blocks interleaved with expressive Transformer layers. These hybrids achieve competitive performance within their training context (e.g., 4K tokens), but their performance beyond training length remains largely untested. Several recent works further explore architectural modifications to extend the context length of such hybrid models, often requiring retraining from scratch. For example, Hybma (Dong et al., 2025) mixes SSM and attention at the head level, while Samba (Ren et al., 2024) replaces full attention with sliding-window attention. Because their long-context behavior is primarily determined by architectural design and training-time context windows, they are not directly comparable to post-hoc context extension methods that operate purely at inference time. In contrast, UPI is designed to extend the context length of existing hybrid Mamba–Transformer models without modifying model architecture or requiring additional training. UPI operates solely at inference time and is therefore directly applicable to off-the-shelf hybrid models.

**Context Length Extension.** We refer to the problem of generalizing models trained on short contexts, to longer sequences as context length extension (CLE). In Transformer models, position encoding plays a central role in CLE. Early approaches use additive positional bias (e.g., T5 (Raffel et al., 2019), ALiBi (Press et al., 2022)), while more recent LLMs (Touvron et al., 2023; AI, 2024; Bai et al., 2023; Qwen et al., 2024) adopt multiplicative positional embeddings such as RoPE (Su et al., 2023). CLE in Transformers has been widely studied through RoPE modifications, including position interpolation (PI) (Chen et al., 2023), NTK-aware scaling (LocalLLaMA, 2023a;b), YaRN (Peng et al., 2024), and LongRoPE (Ding et al., 2024; Shang et al., 2025). These methods rescale positional embeddings across RoPE dimensions to enable interpolation or extrapolation of position information.

CLE methods also exist for Mamba models, such as DeciMamba (Ben-Kish et al., 2025), Long-Mamba (Ye et al., 2025), StuffedMamba (Chen et al., 2025), and MambaExtend (Azizi et al., 2025). These approaches often rely on discrete mechanisms such as token filtering or time step clamping, introducing inference-time overhead or requiring costly tuning on long sequences. Prior work has also focused exclusively on pure Transformer or Mamba architectures. To our knowledge, training-free CLE for hybrid models remains largely unexplored. This gap motivates our work: identifying hybrid model failure modes and proposing a lightweight, closed-form scaling method to address them.

## 3 Understanding Context Limitations in Hybrid Models

Hybrid Mamba–Transformer models perform strongly within their training contexts, but degrade sharply beyond them. As shown in Fig. 1, state-of-the-art hybrid models consistently struggle when evaluated on context lengths unseen during training. Bamba v2 and Nemotron-H (base) experience substantial performance degradation (solid lines): perplexity explodes while retrieval accuracy drops precipitously. A common assumption, seen in works like Hymba (Dong et al., 2025) and Jamba (Lenz et al., 2025), is that Transformer and Mamba layers play complementary roles: attention handles precise retrieval, while SSM summarizes local context. Under this view, hybrid model CLE seems as simple as scaling RoPE in the Transformer layers. However, the dashed lines in Fig. 1 reveal this approach falls short. Applying YaRN yields mild gains, but fails to prevent performance collapse. This motivates a deeper look into why hybrid models fail on long contexts.

### 3.1 Effective Receptive Field (ERF) Analysis

We start by extending effective receptive field (ERF) analysis (Ben-Kish et al., 2025; Ye et al., 2025) to hybrid models, covering both Mamba and Transformer layers. ERF quantifies the influence of each input token on a given output position, illustrating how information propagates across sequence length. A detailed description of ERF is given in Appendix C. We find that within the training context, Mamba layers with the highest average ERF perform comparably to Transformer layers. However, as sequence length increases, ERF of Mamba layers saturates, whereas Transformer layers do not, revealing a weakness in Mamba's ability to scale. Fig. 2 (left) shows highly non-uniform ERF patterns: while the largest values by far come from the transformer layers (L9/18/27), some Mamba layers attend several hundred tokens away in expectation. Averaging across heads also obscures significant variation between them. A fine-grained head- and layer-wise ERF analysis also reveals substantial variability between individual layers and heads. Fig. 2 (right) shows that Mamba heads are highly heterogeneous, with the highest-ERF heads rivaling their transformer counterparts, despite sharing the same update recurrence with heads that saturate quickly. What accounts for this disparity? In § 3.2, we analyze and connect this behavior to hybrid model failure modes on long inputs.

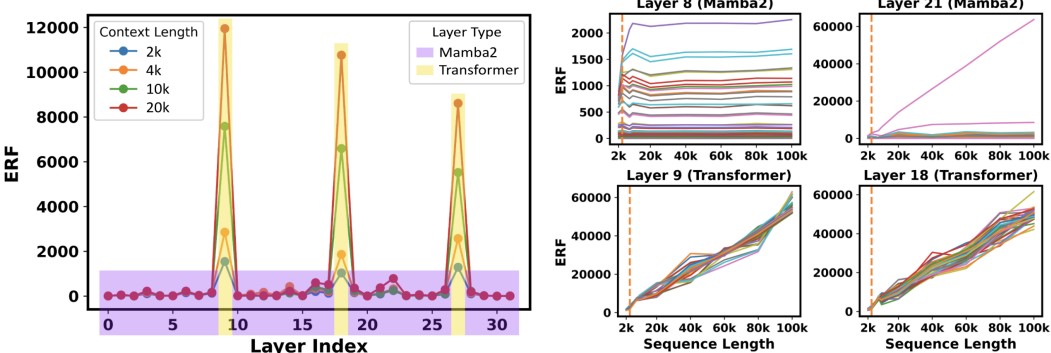

Figure 2: **Left**: ERF distribution across layers of Bamba-9B-v2. For each layer, the ERF is averaged across heads. Yellow highlighted layers (L9, L18, L27) are Transformer layers. **Right**: ERF as a function of context length for all heads in a layer. Each curve represents ERF change w.r.t. sequence length in one head. Transformer heads depict consistent near-linear ERF growth, while most Mamba heads have saturated ERF once sequence length exceeds training context. A few Mamba heads continue to expand ERF comparably to Transformer heads.

## 3.2 STATE MAGNITUDE EXPLOSION IN MAMBA LAYERS

The ERF analysis in § 3.1 reveals that Transformer receptive fields expand reliably with input length, while only some Mamba heads do the same. This imbalance stems from the dynamics of Mamba states themselves: certain heads accumulate state aggressively, leading to uncontrolled growth at longer contexts. To understand this phenomenon, we analyze the Mamba's recurrent update rule.

As introduced in section 2, the update rule for Mamba layer state can be simplified to: $\mathbf{h}_{t+1} = \overline{a}_t \mathbf{h}_t + \overline{\mathbf{B}}_t \mathbf{x}_t$. Note that $\forall t : 0 \leq \overline{a}_t \leq 1$, as $\overline{a}_t$, the forget gate, is sigmoid-activated. Similarly, $\overline{\mathbf{B}}_t$ and $\mathbf{x}_t$ are mostly positive, since $\mathbf{x}_t$ is Silu-activated, and $\overline{\mathbf{B}}_t$ is the product of softplus- and Silu-activated values. For simplicity, let $\overline{\mathbf{B}}_t$ and $\mathbf{x}_t$ be always positive. If we assume that $\overline{a}_t, \overline{\mathbf{B}}_t, \mathbf{x}_t$ are roughly constant over time ($(\overline{a}_t, \overline{\mathbf{B}}_t, \mathbf{x}_t) \approx (\overline{a}, \overline{\mathbf{B}}, \mathbf{x})$), it can be easily shown that the sequence $\mathbf{h}_{0\ldots t}$ forms successive terms of a geometric series:

$$\mathbf{h}_0 = \overline{\mathbf{B}}\mathbf{x}, \quad \mathbf{h}_1 = \overline{a}(\mathbf{h}_0) + \overline{\mathbf{B}}\mathbf{x} = \overline{a}\overline{\mathbf{B}}\mathbf{x} + \overline{\mathbf{B}}\mathbf{x}, \quad \ldots$$

$$\mathbf{h}_t = \sum_{i=0}^{t} \overline{a}^i \overline{\mathbf{B}}\mathbf{x} < \sum_{i=0}^{\infty} \overline{a}^i \overline{\mathbf{B}}\mathbf{x} = \frac{\overline{\mathbf{B}}\mathbf{x}}{1 - \overline{a}}$$

All values are positive and $\overline{a} < 1$, so this series converges asymptotically in the limit to a finite upper bound $\overline{\mathbf{B}}\mathbf{x}/(1 - \overline{a})$. Thus the Frobenius norm of a matrix-valued state $\mathbf{h}$ would also converge to a finite upper bound, since every matrix element converges to its own positive supremum individually. In practice, these terms are not constant or consistently positive, but we still observe this predicted behavior in Mamba layers, as shown in Fig. 3. State magnitude varies noisily due to Mamba's dynamic gating, but it grows and saturates alongside ERF within each head.

This pattern becomes problematic when the forget gate is consistently close to 1. Returning to our simplified model, the asymptotic upper bound for scalar $\mathbf{h}$ is given by $\overline{\mathbf{B}}\mathbf{x}/(1 - \overline{a})$. When $\overline{a} \approx 1$, the series converges extremely slowly: upper bound $\overline{\mathbf{B}}\mathbf{x}/(1 - \overline{a})$ is much larger than the initial (and greatest) summand $\overline{\mathbf{B}}\mathbf{x}$, and the individual summands $\overline{a}^t \overline{\mathbf{B}}\mathbf{x}$ converge to zero only for very large $t$. One can easily envision a Mamba head with forget gate so consistently close to 1 that its state magnitude takes more time to approach its supremum than is available in the training sequence length. And this indeed occurs in practice. Heads in Fig. 3 with low state magnitude reach their upper bounds quickly, then hover as sequence length extends beyond the training length. Meanwhile, a high-ERF head grows far beyond any scale seen during training, as it has yet to converge to its supremum.

Such out-of-distribution growth produces feature collapse in Mamba layers. The output of the SSM operation is given by $\mathbf{y}_t = \mathbf{C}_t \mathbf{h}_t + \mathbf{D}\mathbf{x}_t$, where $\mathbf{y}_t, \mathbf{C}_t, \mathbf{x}_t, \mathbf{D}$ are all vector-valued. Here, $\mathbf{h}_t$ represents the state matrix for every head concatenated in the horizontal dimension. Thus when head $n$ has a state $\mathbf{h}_t^{(n)}$ with very large magnitude, the first term $\mathbf{C}_t \mathbf{h}_t^{(n)}$ overwhelms the second. Likewise,

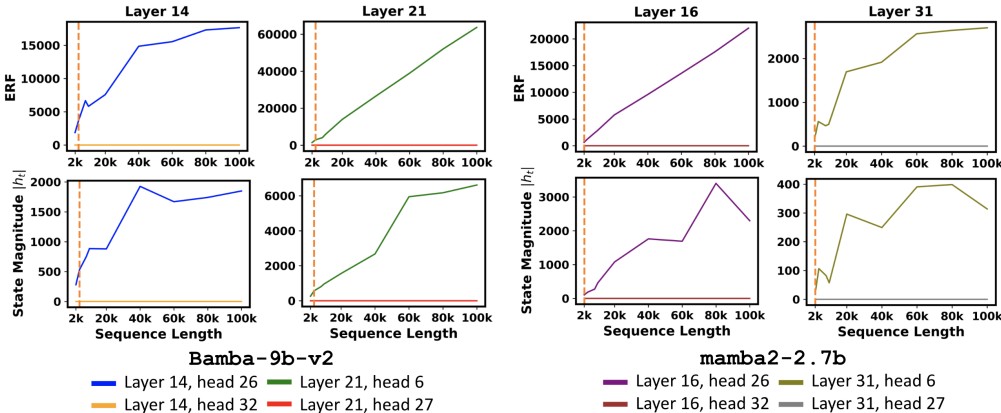

Figure 3: Frobenius norm of hidden state over sequence length for BambaV2 (**Left**) and Mamba2 (**Right**), using a sample from PG-19 validation set. Each color represents a different head. ERF and state magnitude grow together, with high-ERF heads overwhelming saturated heads on long inputs.

the submatrix product $\mathbf{C}_t \mathbf{h}_t^{(n)}$ dominates the concatenated vectors coming from the other heads. The next step in the Mamba layer is a GroupNorm across heads, which scales the large vector terms back into distribution, but in doing so suppresses all terms coming from other heads or inputs. This leads to sparse activations, and the vast majority of layer weights contributing to dead neurons. We claim it is this feature collapse that is responsible for Mamba and hybrid model failure on long contexts.

In order to scale Mamba and hybrid model inputs, we must then control the state magnitude of unconverged heads. One simple approach would be to reduce further the contribution of individual tokens in such heads: for an input $n$ times longer than the training length, we would want each token to shift the state matrix $1/n$ times as much as it otherwise would have. This idea forms the backbone of our approach, which we formalize in the next section.

## 4 UNIVERSAL POSITION INTERPOLATION

### 4.1 MOTIVATION AND INTUITION

The analysis in § 3 reveals that collapse arises from unstable state magnitudes in a subset of Mamba heads. These heads increment their states far beyond the training distribution, suppressing contributions from other heads. A natural question is how to prevent this extra drift. Our intuition is simple: when sequence length is scaled up by a factor $n$, the hidden state contribution of each token should be scaled down by $1/n$. Otherwise, the cumulative state grows disproportionately, producing the observed runaway heads. Phrased differently, the magnitude of unconverged head states can be kept in-distribution by "slowing down" the state's rate of change proportional to the increased input length. This view motivates *Universal Position Interpolation* (UPI), a training-free, closed-form method that rescales token contributions to maintain stable state dynamics as sequence length grows.

UPI is "universal" in two senses. First, it applies to both Mamba and Transformer components of hybrid models, ensuring a consistent treatment of positional dynamics across architectures. Second, it requires no retraining, additional parameters, or structural changes: it is purely a closed-form rescaling that can be inserted at inference time. This makes UPI especially well-suited for hybrid models, where only a subset of heads require correction.

### 4.2 CLOSED-FORM DERIVATION

We now formalize the intuition behind UPI. We start with Mamba's original state update rule:
$$\mathbf{h}_{t+1} = \exp\left(-a\Delta_t\right)\mathbf{h}_t + \Delta_t \mathbf{B}_t \mathbf{x}_t.$$
Let the state update rule on the expanded sequence with scaling factor $n$ take the following form:
$$\mathbf{h}_{t+\frac{k+1}{n}} = f(a, \Delta_t, n)\mathbf{h}_{t+\frac{k}{n}} + g(\mathbf{B}_t, \Delta_t, n)\mathbf{x}_t, \; \forall 0 \leq k < n,$$

where $f(a, \Delta_t, n)$ and $g(\mathbf{B}_t, \Delta_t, n)$ are adjusted forget and input gates. Unrolling the recurrence, and setting the gates equal to the ones from the original update rule, yields adjusted formulas:

$$\exp\left(-a\Delta_t\right) = f(a, \Delta_t, n)^n \qquad \text{(Forget Consistency)}$$

$$\Delta_t \mathbf{B}_t = g(\mathbf{B}_t, \Delta_t, n) \cdot \sum_{k=0}^{n-1} f(a, \Delta_t, n)^k$$

$$= g(\mathbf{B}_t, \Delta_t, n) \cdot \frac{f(a, \Delta_t, n)^n - 1}{f(a, \Delta_t, n) - 1} \qquad \text{(Input Consistency)}$$

Here we are enforcing two consistency conditions: (i) forget consistency: after $n$ small steps, the total forgetting should equal one big step; and (ii) input consistency: input contributions accumulated over $n$ small steps should equal one big step. Solving these produces the following gate functions,

$$\begin{cases} f(a, \Delta_t, n) = \exp\left(-\frac{a}{n}\Delta_t\right), \\ g(\mathbf{B}_t, \Delta_t, n) = \Delta_t \mathbf{B}_t \cdot \frac{1 - \exp\left(-\frac{a}{n}\Delta_t\right)}{1 - \exp(-a\Delta_t)}. \end{cases} \qquad (1)$$

For the forget gate, this amounts to taking the $n$th root, scaling it closer to 1, while the input gate receives an adjusted fraction between 1 and $1/n$ based on the forget gate. In theory, for an input $n$ times longer than the original, the above formula scales the forget- and input-gates of unconverged heads, bringing state magnitudes back into distribution. Unfortunately, this does not actually work in practice. While the provided adjustments scale down state magnitude growth by $\frac{1}{n}$ in expectation, we find that data-dependent input gate scaling behaves too erratically when $\Delta_t$ varies aggressively in practice. Thankfully, this can be solved by a simple tweak to the input gate scale factor.

Recall that these adjustments are applied specifically to Mamba heads with forget gate close to 1 in expectation. From the Mamba update rule, the forget gate approaches 1 as $\Delta_t$ approaches 0. While the data-dependent input gate scaling factor is undefined for $\Delta_t = 0$, we can still take the limit:

$$\lim_{\Delta_t \to 0} \frac{1 - \exp\left(-\frac{a}{n}\Delta_t\right)}{1 - \exp\left(-a\Delta_t\right)} = \frac{1}{n}. \qquad (2)$$

The input gate scaling factor can thus be replaced by constant $1/n$. This reduces further to the original Mamba update rule, with $\Delta_t$ divided by $n$. Thus for an input $n$ times longer than the training length, we can simply divide $\Delta_t$ by $n$ for unconverged heads and attain good performance, as shown in § 5.1.

## 4.3 STATE SPACE DUALITY AND POSITIONAL ENCODING

So far we have motivated UPI empirically: it stabilizes unstable heads and restores balanced context scaling. But is this simply a heuristic, or does it connect to deeper principles of sequence modeling? To answer this, we revisit the structured state space duality between Mamba and Transformers (Dao & Gu, 2024), which posits a component-wise correspondence across the two architectures: Transformer's query, key, and value correspond to Mamba's output gate $\mathbf{C}_t$, input gate $\overline{\mathbf{B}}_t$, and input token $\mathbf{x}_t$, respectively. Similar to multi-head attention, input and output matrices $\mathbf{B}_t$ and $\mathbf{C}_t$ are shared across heads, while forget gate and step size use head-specific parameters $a$ and $\Delta_t$.

Building on this foundation, we make a further observation: Mamba's compounding forget gates and Transformer's RoPE encodings both act as mechanisms for injecting position into the model. From this lens, UPI can be viewed as a principled extension of position interpolation: both determine how representations advance with token position, and gate scaling in Mamba is mathematically analogous to frequency scaling in RoPE (Su et al., 2023). In RoPE, interpolation methods (Chen et al., 2023) such as YaRN rescale the lowest frequencies to enable longer contexts; in Mamba, UPI similarly rescales small $\Delta_t$. This conceptual bridge explains why UPI succeeds on hybrid models, and why it can be seen as the first training-free CLE tailored to them.

We provide a proof sketch of this equivalence in Appendix E. Both RoPE and Mamba maintain a bank of keys, and multiply key $j$ for input at time $n$ by $\exp\left(\sum_{t=j}^{n} \Delta_t\right)$, with $\Delta_t$ varying per-channel(head) and position. In RoPE, $\Delta$ is a static complex constant (w.r.t. $t$); in Mamba, $\Delta_t$ is real-valued and input-dependent. Both can be manipulated analogously: YaRN scales small $\Delta_t$ (low frequencies) by $1/s$, skips large ones, and interpolates between. UPI is the state-space analogue, scaling small Mamba $\Delta_t$ by $1/s$ and leaving the rest untouched. Consistent initializations and similar long-context failure modes further support this RoPE–Mamba duality, with further details in Appendix E and F.

These intriguing links provide avenues for further exploration, but we leave this for future work and simply indicate the duality as additional justification for UPI. In this sense, our work establishes a principled bridge between Transformer position encoding and Mamba gate dynamics.

### 4.4 APPLYING UPI ACROSS HYBRID MODELS

Having established both the intuition and theoretical grounding of UPI, the last outstanding question is how to identify unconverged heads in a model. Unlike RoPE, Mamba $\Delta_t$ values are dynamically generated, not static, and thus unknown *a priori*. While it is possible to analyze state growth per head in detail, and define thresholds for when state magnitude has approached its supremum sufficiently closely within the training sequence length, we find it easier to re-use existing code for calculating ERF, and select the top-$K\%$ of highest-ERF heads for adjustment[1]. As shown in section 3.2, heads with large ERF tend to have large final state magnitude, given that both are determined by the average forget gate value. ERF profiles are highly stable, as shown in § 5.2, and only a few parallel forward passes on a small calibration dataset are required to reliably identify the top-$K\%$ of heads by ERF.

The final UPI procedure is lightweight and general. Broadly, UPI works by identifying components prone to long-context instability, and applying the appropriate closed-form scaling rule from § 4.2.

---

**UPI Pipeline**

**Step 1**: Choose an existing RoPE frequency adjustment technique for Transformer layers

**Step 2**: Take a few forward passes to identify unconverged Mamba heads (top-$K\%$ by ERF)

**Step 3**: Adjust Transformer layer RoPE frequencies, and selected Mamba heads, to match the length of each given input

---

The result is a robust, tuning-free approach to context length extension that incurs no additional inference time overhead, minimal architectural intervention, and little extra computation to perform the model adjustment (a few forward passes to find the unconverged heads). In the following section, we validate and analyze these claims via multiple long-context benchmarks and experiments.

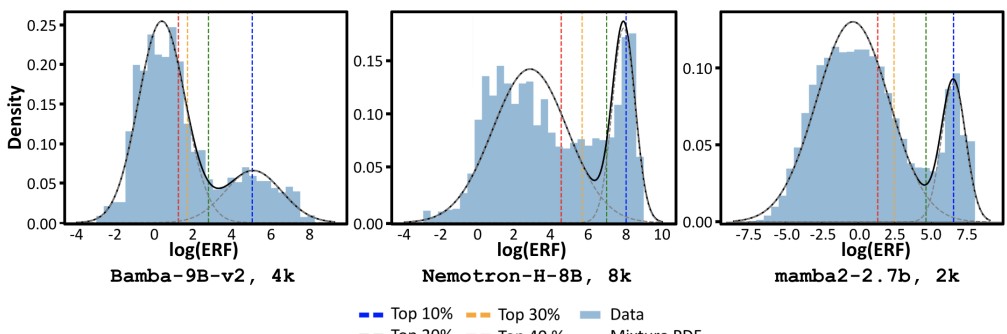

Figure 4: Distribution of Log-ERF across Mamba heads in different models at their training context length: Bamba-9B-v2 at 4k, Nemotron-H-8B at 8k, and Mamba2-2.7B at 2k. The distributions of all three models show bimodal characteristic, motivating our head-wise selective mechanism.

To justify our head-wise selective mechanism, we examine the distribution of the profiled ERF on each heads for all three models, Bamba-9B-v2, Nemotron-H-8B, and Mamba2-2.7B, as presented in Fig. 4. The top-$K\%$ cutoff is motivated by the empirical distribution of log-ERF scores across heads. In all evaluated models, the log-ERF distribution is bimodal: a minority subset of high-ERF heads (approximately $10 \sim 30\%$, depending on layer and sequence length) separates cleanly from the majority of low-ERF heads. Exact percentages vary by model, but we adopt 20% as a general and effective cutoff because it closely aligns with the average split point of the bimodal log-ERF distribution and is further supported by the empirical results in Table 5.

---

[1]Alternative metrics for head selection are discussed in Appendix G.

Table 1: Perplexity on PG-19 at different sequence lengths across different hybrid models.

| Model | Method | 2K | 4K | 8K | 16K | 32K | 64K |
|-------|--------|-----|-----|-----|------|------|------|
| | Base | 8.35 | 8.78 | 9.37 | 14.23 | 32.76 | 127.90 |
| Bamba-9B-v2 | LongMamba + YaRN | 8.42 | 8.51 | **8.83** | 13.02 | 23.68 | 53.14 |
| | **UPI(ours) + YaRN** | **8.13** | **8.30** | 9.01 | **9.82** | **14.60** | **18.59** |
| | Base | 7.53 | 7.82 | 7.59 | 46.57 | 210.30 | 530.31 |
| Nemotron-H-8B | LongMamba | 7.82 | **7.71** | **7.13** | 23.24 | 46.42 | 78.51 |
| | **UPI(ours)** | **7.46** | 7.91 | 7.39 | **15.58** | **25.72** | **44.01** |
| | Base | 7.52 | 8.92 | 16.73 | 53.14 | 137.82 | 478.21 |
| Mamba2-2.7B | LongMamba | **7.31** | **7.94** | 9.16 | 14.82 | 23.59 | 42.20 |
| | **UPI(ours)** | 7.40 | 8.21 | **8.75** | **13.69** | **17.58** | **22.24** |

## 5 EXPERIMENTS

**Models and Benchmarks.** Our main experiments evaluate UPI on a Mamba2 model (`mamba2-2.7B` (Dao & Gu, 2024)) and two hybrid models (`Bamba-9B-v2` (Ganti et al.) and `Nemotron-H-8B-Base-8K` (NVIDIA et al., 2025b)). The Mamba2 model is trained with a 2k context length, Bamba with 4k, and Nemotron-H with 8k. For language modeling capability, we evaluate perplexity on the validation set of PG-19 (Rae et al., 2019). Shorter-than-expected sequences are concatenated until they exceed the target length, and overflow is discarded. LongBench-E (Bai et al., 2024) and RULER (Hsieh et al., 2024a) are used to evaluate long-range understanding.

**Baselines.** Since no existing tuning-free CLE methods have been developed for hybrid models, we compare against component-wise CLE baselines, applying separate strategies to Mamba and Transformer layers. For Transformer layers with RoPE, we adopt YaRN (Peng et al., 2024), a tuning-free position interpolation method. For Mamba layers, we compare our proposed UPI method against LongMamba (Ye et al., 2025), a strong CLE baseline that, while tuning-free in architecture, requires parameter search for optimal performance. Hyperparameters for LongMamba are optimized over several rounds of LongBench-E, which take roughly 10 hours on 8 A100 GPUs for each model. UPI is calibrated by running forward passes on samples from the ArXiv subset of proof-pile (Computer, 2023), which takes under 3 minutes. The cost during inference is negligible for both methods. Further implementation details are provided in Appendix A.

### 5.1 MAIN RESULTS: EVALUATING HYBRID MODELS ON LONG-CONTEXT TASKS

**PG-19 Perplexity.** As shown in Table 1, UPI excels at extending hybrid models to long contexts without fine-tuning. It outperforms not only base models that degrade rapidly, but also the strong LongMamba + YaRN baseline, which still struggles to maintain low perplexity at extended lengths. The gains are substantial: at 64k on Bamba, UPI reaches a perplexity of 18.59—an $85.5\%$ relative reduction vs. the base model; the relative reductions on Nemotron-H and Mamba-2 are $91.7\%$ and $95.3\%$, respectively. This result reflects UPI's theoretical grounding: state behavior is kept in-distribution as promised, even when inputs are much longer than expected.

**RULER.** We further evaluate UPI on the RULER benchmark for long-context retrieval and reasoning tasks across various domains. As shown in Table 2, UPI + YaRN consistently outperforms both base hybrid models and the LongMamba + YaRN baseline for all extended context lengths. The method excels at retrieval (S1-S3, M1-M3, MV, MQ), especially on tasks like S2 where it outperforms competitors by a substantial margin. While it is not immune to the expected performance degradation on complex reasoning tasks (VT, CWE, FWE, QA1, QA2), UPI + YaRN mitigates this drop effectively. Moreover, UPI also demonstrates similar superior performance when applied to the pure Mamba2 model, proving its robustness over both hybrid and pure Mamba architectures.

**LongBench.** We also report results on LongBench-E in Table 3, which targets long-range dependency. UPI + YaRN consistently surpasses both baselines in both hybrid models on single/multi-document QA and summarization, and overall, with a small dip on few-shot tasks. We note that while UPI trails LongMamba when applied to Mamba2, this is likely a feature of LongMamba rather than UPI. LongMamba is optimized end-to-end directly for this task family, while UPI represents a general closed-form solution. In hybrid models, there is no LongMamba equivalent for transformer layers,

Table 2: Comparison of performance (%) on RULER with extended context lengths.

| Model | Len | Method | S1 | S2 | S3 | M1 | M2 | M3 | MV | MQ | VT | CWE | FWE | QA1 | QA2 | Avg. |
|---|---|---|---|---|---|---|---|---|---|---|---|---|---|---|---|---|
| Bamba-9B-v2 | 8k | Base | 25.6 | 35.8 | 38.6 | 25.6 | **16.8** | 0.6 | 27.8 | 27.5 | 7.1 | 1.2 | 11.7 | 26 | 17.6 | 20.1 |
| | | LongMamba + YaRN | 49.6 | 67.0 | **63.0** | 22.0 | 16.6 | 1.6 | 27.5 | 27.6 | **11.2** | 0.8 | 8.7 | 24.0 | 18.8 | 26.0 |
| | | **UPI(ours) + YaRN** | 55.8 | 68.0 | 49.2 | 27.0 | 15.4 | 0.6 | 31.9 | 31.8 | 4.7 | 3.3 | 26.4 | 30.8 | 22.8 | 28.3 |
| | 16k | Base | 43.6 | 16.2 | 24.8 | 11.6 | 0.0 | **0.0** | 8.0 | 3.9 | 1.3 | **9.7** | **0.9** | 16.4 | 20 | 12.0 |
| | | LongMamba + YaRN | **72.8** | 20.6 | 0.4 | 13.2 | 0.2 | 0.0 | 9.2 | 7.3 | 0.0 | 2.3 | 0.3 | 9.2 | 13.0 | 11.4 |
| | | **UPI(ours) + YaRN** | 60.6 | 40.6 | 34.8 | 21.8 | 0.6 | 0.0 | 19.1 | 15.0 | 5.0 | 6.2 | 0.3 | 22.6 | 20.8 | 19.0 |
| Nemotron-H-8B | 16k | Base | 54.8 | 63.1 | 31.8 | **38.8** | 45.1 | 12.0 | 54.8 | 33.3 | 78.2 | 57.4 | 42.1 | 4.2 | 13.4 | 40.7 |
| | | LongMamba | **67.3** | 42.8 | 35.1 | 30.0 | 36.7 | 8.7 | **62.3** | 25.4 | **84.4** | 73.9 | **52.1** | 6.7 | **24.9** | 41.9 |
| | | **UPI(ours)** | 65.8 | 67.2 | 36.9 | 24.8 | 52.0 | 4.3 | 47.9 | 43.8 | 82.0 | 88.5 | 46.5 | 15.7 | 8.7 | 45.0 |
| | 32k | Base | 26.3 | 17.2 | 11.1 | 16.1 | 4.9 | **0.0** | 44.1 | 8.7 | **63.9** | 43.2 | **18.6** | 0.0 | **17.8** | 20.9 |
| | | LongMamba | 34.7 | 12.0 | 3.8 | 8.5 | 0.4 | 0.0 | 45.3 | 11.8 | 59.0 | 40.5 | 8.8 | 0.9 | 14.3 | 18.5 |
| | | **UPI(ours)** | 45.1 | 32.2 | 35.6 | 21.4 | 7.2 | 0.0 | 52.2 | 11.0 | 61.4 | 46.6 | 14.7 | 3.2 | 7.8 | 26.0 |
| Mamba2-2.7B | 4k | Base | 100.0 | 0.0 | 0.0 | 0.0 | **0.6** | 0.4 | 0.0 | 0.0 | **48.7** | 3.9 | 66.6 | **37.6** | 19.0 | 21.3 |
| | | LongMamba | 100.0 | 3.4 | 1.8 | 0.2 | 0.0 | 0.0 | 1.7 | 1.3 | 4.7 | **9.0** | 31.2 | 36.6 | **27.5** | 16.7 |
| | | **UPI(ours)** | 100.0 | 10.4 | 4.4 | 8.6 | 0.0 | 0.0 | 2.9 | 1.45 | 37.2 | 2.5 | 71.1 | 29.8 | 11.6 | 21.5 |
| | 8k | Base | 99.2 | 0.0 | 0.0 | 0.0 | **0.0** | 0.0 | 0.0 | 0.0 | 17.7 | 0.0 | 0.5 | 0.2 | 0.2 | 9.1 |
| | | LongMamba | 100.0 | 2.3 | 1.6 | 4.3 | 0.0 | 0.0 | 0.8 | 0.6 | 2.3 | **3.5** | 1.3 | **8.8** | **13.0** | 10.7 |
| | | **UPI(ours)** | 100.0 | 5.8 | 2.2 | 5.2 | 0.0 | 0.0 | 2.0 | 1.3 | 31.4 | 0.5 | 11.6 | 8.0 | 9.4 | 13.6 |

so Mamba and transformer scaling cannot be optimized jointly, and the benefits of LongMamba evaporate. UPI, meanwhile, is consistent in boosting performance for long contexts.

Table 3: LongBench-E results on different category of tasks.

| Model | Method | SQA | MQA | Summary | Few-shot | Synthetic | Code | Avg. |
|---|---|---|---|---|---|---|---|---|
| Bamba-9B-v2 | Base | 11.68 | 10.09 | 18.69 | 49.19 | **6.64** | 38.69 | 21.71 |
| | LongMamba + YaRN | 12.16 | 8.43 | 17.49 | **56.57** | 2.53 | **46.85** | 22.85 |
| | **UPI(ours) + YaRN** | **12.20** | **13.25** | **20.94** | 52.71 | 5.85 | 41.68 | **23.68** |
| Nemotron-H-8B | Base | 14.87 | 12.36 | 19.74 | 50.36 | 8.24 | 40.89 | 23.61 |
| | LongMamba | 16.47 | 12.29 | 17.54 | **51.92** | 7.65 | 37.03 | 23.14 |
| | **UPI(ours)** | **16.81** | **14.48** | **21.86** | 48.8 | **8.47** | **46.85** | **25.15** |
| Mamba2-2.7B | Base | 8.79 | 3.95 | 9.04 | 10.78 | 2.32 | 22.89 | 9.72 |
| | LongMamba | 13.07 | **9.76** | **14.95** | **16.54** | **10.50** | **29.40** | **15.77** |
| | **UPI(ours)** | **13.60** | 6.47 | 13.23 | 15.58 | 8.50 | 29.21 | 14.52 |

## 5.2 ANALYSIS RESULTS

**How important is scaling both Mamba and Transformer layers?** We perform an ablation study of UPI on PG-19 perplexity by selectively disabling UPI in either the Transformer or Mamba layers of Bamba-v2. As shown in Table 4, removing either part degrades performance. Notably, disabling UPI at Mamba layers produces higher perplexity than disabling YaRN, which aligns with the ERF analysis in 3 showing that Mamba layers are the bottleneck for context scaling of hybrid models. Moreover, if we interpolate all Mamba heads, performance likewise drops dramatically, echoing our findings that particular unconverged heads are the issue.

Table 4: Ablation study on UPI components.

| Method | 8K | 16K | 32K | 64K |
|---|---|---|---|---|
| UPI + YaRN | 9.01 | 9.82 | 14.60 | 18.59 |
| w/o YaRN | 9.23 | 12.54 | 16.87 | 25.79 |
| w/o UPI | 8.97 | 14.85 | 27.29 | 98.01 |
| w/o Selective head | 9.47 | 15.73 | 23.85 | 65.92 |

Table 5: Ablation study on the Top-K setting.

| Top-K | 8K | 16K | 32K | 64k |
|---|---|---|---|---|
| 10% | 10.43 | 18.79 | 24.21 | 33.69 |
| 20% | 9.01 | 9.82 | 14.60 | 18.59 |
| 30% | 12.18 | 21.09 | 20.14 | 27.02 |
| 40% | 16.24 | 22.76 | 22.53 | 34.14 |

**Is it always beneficial to scale more heads?** We perform further ablations on PG-19 perplexity by testing different filtering thresholds for unconverged heads in Bamba-v2. As shown in Table 5, setting the threshold to 20% yields the best performance across context lengths. Conducting interpolation on more than 30% of heads causes performance to degrade noticeably, indicating that aggressive interpolation disrupts short-term heads, harming ability to model local dependencies.

**Is ERF-based head selection susceptible to dataset change?** We compare the top-K head selections calculated from the validation split of three calibration datasets: proof-pile ArXiv subset (Computer, 2023) (D1), PG-19 (Rae et al., 2019) (D2), and wikitext2-v1 (Merity et al., 2016) (D3). As shown in Table 6, the overlap increases as the top-K percentage decreases, indicating that the heads with

the highest ERF are consistent across datasets. For our top-20% filter, the overlap between datasets is consistently larger than 80%, indicating fairly robust selections. Thus, using a reasonably small calibration dataset (such as the proof-pile ArXiv subset, with only 100 samples) we can rank the ERF of the heads with confidence.

**UPI achieves fast calibration process.** As decribed in section 4.4, for per-model parameter selection, our pipeline (i) applies YaRN to Transformer layers with RoPE, (ii) profiles ERF for Mamba heads, and (iii) applies a constant scaling to the top 20% heads by ERF. Stages (i) and (iii) add negligible overhead. For stage (ii), we calibrate per-head ERF on 100 16k-token sequences that completes in $<$ **3 minutes** on 8 A100 GPUs. By contrast, LongMamba tunes hyperparameters via multiple LongBench runs, taking $\sim$ **10.5 hours** on the same hardware. This gap underscores the lightweightness of our approach. At inference, both methods apply element-wise operations over the $\Delta_t$ array, yielding no extra overhead. See Appendix A for details.

Table 6: Ablation study on head selection overlap percentage (%) across different datasets.

| Top-K | D1–D2 | D2–D3 | D1–D3 |
|---|---|---|---|
| 10% | 91.8 | 95.2 | 90.4 |
| 20% | 82.7 | 91.0 | 87.5 |
| 30% | 75.2 | 80.9 | 68.1 |
| 40% | 57.2 | 67.3 | 48.1 |

**Scalability and Generalizability** To demonstrate the generalizability of our method to larger models and diverse tasks, we evaluate Nemotron-H-47B-Base-8k on the Dialogue History QA and Code Repo QA tasks from LongBench v2. The results are shown in Table 7. Our method consistently improves performance on retrieval, dialogue understanding, and code repository understanding for the 47B model, indicating strong generalization.

| Model | Method | Passkey Retrieval | | | Dialogue | Code |
|---|---|---|---|---|---|---|
| | | 8k | 16k | 32k | History QA | Repo QA |
| Nemotron-H | Baseline | **100.0** | 9.6 | 0.0 | 31.58 | 28.00 |
| -47B-Base-8k | UPI | 100.0 | **16.3** | **8.5** | **33.28** | **34.15** |

Table 7: Evaluation of Nemotron-H-47B-Base-8k on Passkey Retrieval from RULER, and Dialogue History QA, and code Repo QA from LongBench v2.

**Complementarity with Fine-Tuning-Based Extrapolation** To illustrate complementarity of UPI and extrapolation methods, we fine-tuned Bamba-9B-v2 to 32k and 128k context lengths. As shown in Table 8, applying UPI to the 32k model improves long-context performance beyond 32k, closing part of the gap to the 128k fine-tuned checkpoint. Additionally, UPI also helps improve the 128k checkpoint on long contexts. This demonstrates that UPI can still provide benefits even when fine-tuning-based extrapolation is available.

| Model | Method | 4k | 8k | 16k | 32k | 64k | 128k |
|---|---|---|---|---|---|---|---|
| Bamba-9B-v2 32k | Base | 84.74 | 78.05 | 70.62 | 62.10 | 35.49 | 18.04 |
| | UPI + YaRN | **85.57** | **79.70** | **71.56** | **63.79** | **40.90** | **28.42** |
| Bamba-9B-v2 128k | Base | **83.93** | **77.30** | 74.12 | 69.35 | 64.36 | 61.57 |
| | UPI + YaRN | 81.85 | 76.57 | **75.62** | **71.43** | **66.91** | **62.17** |

Table 8: Evaluation of Mamba2-2.7B on PG-19 perplexity, comparing the effectiveness of DeciMamba and UPI. DeciMamba's token-pruning scheme shows better performance on short context (2k to 8k), while UPI's state growth stabilization improve performance on even longer context.

## 6 CONCLUSION

We present the first in-depth analysis of hybrid Mamba-Transformer model scaling, showing that long-context failures arise from a small subset of unstable heads. By viewing this through the lens of position encoding and structured state space duality, we introduce Universal Position Interpolation (UPI), a closed-form and tuning-free method for context scaling. UPI extends hybrid models far beyond their training context lengths while preserving short-context accuracy, establishing the first principled bridge between Transformer encodings and Mamba gate dynamics. We hope this work inspires further exploration of training-free CLE methods for hybrid models.

## ACKNOWLEDGMENTS

We sincerely appreciate the anonymous reviewers. Their insightful feedback helps significantly improve the quality of the paper. This research was supported by the National Science Foundation (NSF) under Grant No. 2441601. The work utilized the Delta and DeltaAI system at the National Center for Supercomputing Applications (NCSA) and Jetstream2 at Indiana University through allocation CIS240055 from the Advanced Cyberinfrastructure Coordination Ecosystem: Services & Support (ACCESS) program, which is supported by National Science Foundation grants #2138259, #2138286, #2138307, #2137603, and #2138296. The Delta advanced computing resource is a collaborative effort between the University of Illinois Urbana-Champaign and NCSA, supported by the NSF (award OAC 2005572) and the State of Illinois. UIUC SSAIL Lab is supported by research funding and gift from IBM, Google, Amazon, and AMD.

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

# A  EXPERIMENT DETAILS

**Model Checkpoints.** The model checkpoints we experimented on, as listed in HuggingFace, are:

- `ibm-ai-platform/Bamba-9B-v2` (Ganti et al.): trained on 4k context length, consists of 29 Mamba2 layers and 3 Transformer layers with RoPE.
- `nvidia/Nemotron-H-8B-Base-8K` (NVIDIA et al., 2025b): trained on 8k context length, consists of 24 Mamba2 layers and 4 Transformer layers without RoPE. Note that we are using the 8k base model instead of the one fine-tuned to 128k context length.
- `state-spaces/mamba2-2.7b` (Dao & Gu, 2024): trained on 2k context length, consists of 64 Mamba2 layers.

**Evaluation.** To evaluate the model's long context behavior, we adopt the following three widely used benchmarks

- Perplexity on PG-19 (Rae et al., 2019) test set: consists of 100 books with an average length of 69k tokens.
- RULER (Hsieh et al., 2024a): consists of 8 variants of needle-in-a-haystack retrieval task, and 5 other tasks that test the model's context understanding ability. Note that S1 is equivalent to the passkey retrieval task (Mohtashami & Jaggi, 2023).
- LongBench-E (Bai et al., 2024): consists of 4 tasks on single-document QA, 4 tasks on multi-document QA, 4 tasks on summarization, 4 tasks on few-shot learning, 3 tasks on synthetic tasks, and 2 tasks on code completion. In section § 5.1, we reported the average score under each of the categories, as well as the overall average.

**Baselines.** At the time of writing, the LongMamba (Ye et al., 2025) codebase has not published their pipeline for parameter searching, so we followed the description to implement their two-stage method.

1. Parameter search: Conduct 16 runs on LongBench-E and choose the hyperparameter setting (long-term head threshold $\theta$, and large $\Delta$ clipping percentage $C$) that maximizes the overall average score. This procedure yields superior performance on LongBench-E but incurs a substantial search cost ($\sim$ 10 hours on 8 A100 GPUs) and must be repeated for each new model. We also observe that performance on other benchmarks is sensitive to this choice, making the search overhead necessary.

2. Small $\Delta$ clipping value look-up table: Build a look-up table for the clipping value per 1000 tokens, where each entry is computed via taking inference on 5 samples. We only compute the clipping value for the sequence length we experimented on, so the cost for this phase is 30 forward passes, which is negligible comparing to the first phase.

During inference, LongMamba is efficient as the running time is still dominated by the state space model. In practice, the overall running time of LongMamba's pipeline takes $\sim$ 10.5 hrs on 8 A100 GPUs.

**Universal Position Interpolation (our method).** As described in section § 4.4, our method consists of three steps, where the first and the third step does not require any computation. For the second step, we utilize the validation set of the proof-pile dataset's ArXiv subset (Computer, 2023), consisting of 100 samples. We concatenate and truncate sequences to make sure all inputs contain 16k tokens. The model performs inference on all samples to acquire the average ERF of each Mamba head. This process takes 100 forward passes that can be conducted in parallel, which is less than 3 minutes on 8×A100.

# B  STATE SPACE MODELS AND MAMBA

State Space Models (SSMs) offer a promising alternative to attention-based architectures for processing long sequences efficiently. An SSM is typically defined by a continuous-time dynamical system that captures temporal dependencies through a set of linear differential equations. Formally, an SSM

can be represented as follows:

$$\frac{\mathrm{d}\mathbf{h}(k)}{\mathrm{d}k} = \mathbf{A}\mathbf{h}(k) + \mathbf{B}\mathbf{x}(k), \tag{3}$$

$$\mathbf{y}(k) = \mathbf{C}\mathbf{h}(k) + \mathbf{D}\mathbf{x}(k), \tag{4}$$

where $\mathbf{h}(k)$ represents the hidden state at time $k$, $\mathbf{x}(k)$ the input signal, and $\mathbf{y}(k)$ the output signal. Matrices $\mathbf{A}, \mathbf{B}, \mathbf{C}, \mathbf{D}$ parameterize the system dynamics and are learned during training. To accommodate the model with discrete signals such as natural language, this continuous-time formulation is discretized into a linear recurrent equation:

$$\mathbf{h}_{t+1} = \overline{\mathbf{A}}\mathbf{h}_t + \overline{\mathbf{B}}\mathbf{x}_t, \tag{5}$$

$$\mathbf{y}_t = \overline{\mathbf{C}}\mathbf{h}_t + \overline{\mathbf{D}}\mathbf{x}_t, \tag{6}$$

with discrete-time system matrices $\overline{\mathbf{A}}, \overline{\mathbf{B}}, \overline{\mathbf{C}}, \overline{\mathbf{D}}$. This system can process long sequences efficiently, as they avoid attention's quadratic computation complexity and rely on parallelizable recurrence updates.

Mamba builds on SSMs but adds data-dependent gates to enhance flexibility and expressiveness. The detailed design of Mamba and Mamba2 model can be found in (Gu & Dao, 2023; Dao & Gu, 2024). In this work, we mainly invesitgate context extension methods without modifications to the model architecture.

The basic design of Mamba starts from an SSM discretized by zeroth-order hold (ZOH), while making the system matrices context-aware:

$$\mathbf{h}_{t+1} = \overline{\mathbf{A}}_t\mathbf{h}_t + \overline{\mathbf{B}}_t\mathbf{x}_t, \tag{7}$$

$$\text{where } \overline{\mathbf{A}}_t := \exp(\Delta_t\mathbf{A}), \tag{8}$$

$$\overline{\mathbf{B}}_t := (\Delta_t\mathbf{A})^{-1} \left(\exp(\Delta_t\mathbf{A}) - \mathbf{I}\right) \cdot \Delta_t\mathbf{B}_t, \tag{9}$$

$$\Delta_t := \text{SoftPlus}(\mathbf{W}_\Delta\mathbf{x}_t + \mathbf{b}_\Delta), \tag{10}$$

$$\mathbf{B}_t := \mathbf{W}_\mathbf{B}\mathbf{x}_t \tag{11}$$

where all subscripts denote the dependency on the current input $\mathbf{x}_t$, and $\Delta_t$ denotes the discretization time step of it. By letting the matrices vary with the current input $\mathbf{x}_t$, Mamba implements a selective gating mechanism—akin to RNN forget and input gates—that dynamically filters and propagates content along the sequence.

In Mamba2, the design is further simplified as the following,

- $\mathbf{A} := a\mathbf{I}$: The continuous forget gate matrix is simplified to a single parameter for each head.
- $\overline{\mathbf{B}}_t \approx \Delta_t\mathbf{B}_t$: The ZOH discretization rule for the input gate is simplified to a first-order approximation that effectively reduces computation.

By substituting $\Delta_t$ in the expressions of $\overline{a}_t$ and $\overline{\mathbf{B}}_t$, we can obtain the new expressions as:

$$\overline{a}_t = \text{Sigmoid}(-(\mathbf{W}_\Delta\mathbf{x}_t + \mathbf{b}_\Delta))^a, \quad \overline{\mathbf{B}}_t = \text{SoftPlus}(\mathbf{W}_\Delta\mathbf{x}_t + \mathbf{b}_\Delta) \cdot (\mathbf{W}_\mathbf{B}\mathbf{x}_t), \tag{12}$$

Thanks to the linear updating rules of the Mamba layer, we can unroll the recurrence into the following multiplication,

$$\mathbf{Y} = \mathbf{M}\mathbf{X} \tag{13}$$

$$\mathbf{M}_{i,j} := \begin{cases} \mathbf{C}_i^\top\overline{\mathbf{B}}_i, & \text{if } i = j, \\ \mathbf{C}_i^\top \left(\prod_{k=j+1}^i \overline{a}_k\right) \overline{\mathbf{B}}_j, & \text{if } i > j, \\ 0, & \text{otherwise.} \end{cases} \tag{14}$$

where $\overline{a}_t$ is the *forget gate*, controlling the proportion of information retained to the next step, while $\overline{\mathbf{B}}_t$ serves as an *input gate*, determining how much information is added by the new input.

$\mathbf{M}$ can be unrolled into a structured lower triangular matrix,

$$\mathbf{M} = \begin{bmatrix} \mathbf{C}_0^\top\overline{\mathbf{B}}_0 & 0 & \cdots & 0 \\ \mathbf{C}_1^\top\overline{a}_1\overline{\mathbf{B}}_0 & \mathbf{C}_1^\top\overline{\mathbf{B}}_1 & \cdots & 0 \\ \vdots & \vdots & \ddots & \vdots \\ \mathbf{C}_{n-1}^\top \left(\prod_{k=1}^{n-1} \overline{a}_k\right) \overline{\mathbf{B}}_0 & \mathbf{C}_{n-1}^\top \left(\prod_{k=2}^{n-1} \overline{a}_k\right) \overline{\mathbf{B}}_1 & \cdots & \mathbf{C}_{n-1}^\top\overline{\mathbf{B}}_{n-1} \end{bmatrix}, \tag{15}$$

which mimics the *causal attention map* of a transformer layer. The diagonal entries reflect direct influence from current input, and the lower triangle indicates cumulative influence of prior inputs decayed by forget gates. The outputs are then expressed as a weighted sum of all inputs via this pseudo-attention map.

## C    EFFECTIVE RECEPTIVE FIELD

Effective Receptive Field (ERF) was first introduced for analyzing CNNs (Luo et al., 2016), and later adapted to Transformers (Dosovitskiy et al., 2021) and very recently Mamba (Ben-Kish et al., 2025). Although modern LLMs have theoretical full-context receptive fields (Lecun et al., 1998), the ERF captures the practical extent of this influence. To estimate ERF in our analysis, we adopt the Mamba Mean Distance (MMD) metric introduced by (Ben-Kish et al., 2025) given as:

$$
\text{MMD} \;=\; \mathbb{E}_{j \leq L}\big[d(j, L)\big] = \sum_{j=1}^{L} (L - j) \, \frac{|\mathbf{M}_{L,j}|}{\sum_{i=1}^{L} |\mathbf{M}_{L,i}|},
$$

This measures the expected distance (in token positions) between the last token and the weighted influence of earlier tokens, derived from the unrolled recurrence matrix $\mathbf{M}$. A high MMD indicates long-term memory, i.e., the last token is influenced by tokens far back in the sequence, while a low value indicates short-range behavior, i.e., the model is mostly influenced by recent tokens.

A noticeable advantage of using MMD as our ERF metric is that it operates on the attention score, which means it is also applicable to analyze the behavior of Transformer models. This duality allows a fair comparison between Mamba and Transformer layers, aligning with our goal of investigating hybrid models.

## D    RoPE AND POSITION INTERPOLATION

Rotary Position Embedding (RoPE, Su et al. (2023)) is an increasingly popular method to provide position information into transformer models. Given a position index $m$, an array of frequencies $\Theta = [\theta_0, \theta_1, \ldots, \theta_{d/2-1}]$ and an embedding vector $\mathbf{x} := [x_0, x_1, \ldots, x_{d-1}]^\top$, where $d$ is the dimension of the attention head, RoPE defines a vector-valued complex function $\mathbf{r}(\mathbf{x}, m, \Theta)$ as follows

$$
\mathbf{r}(\mathbf{x}, m, \Theta) = [r(x_0 + \mathrm{i}x_1, m, \theta_0), r(x_2 + \mathrm{i}x_3, m, \theta_1), \ldots, r(x_{d-2} + \mathrm{i}x_{d-1}, m, \theta_{d/2-1})]^\top
$$

$$
= \left[(x_0 + \mathrm{i}x_1)e^{\mathrm{i}m\theta_0}, (x_2 + \mathrm{i}x_3)e^{\mathrm{i}m\theta_1}, \ldots, (x_{d-2} + \mathrm{i}x_{d-1})e^{\mathrm{i}m\theta_{d/2-1}}\right]^\top,
$$

where $\mathrm{i} := \sqrt{-1}$ is the imaginary unit and $\theta_j = f_b^{-2j/d}$, with a fixed predefined base frequency $f_b$ that is commonly set to 10,000 or 500,000.

Position Interpolation (PI, Chen et al. (2023)) defines an attention score aiming for interpolation, whose RoPE $\mathbf{r}$ is replaced by the following $\mathbf{r}'$

$$
\mathbf{r}'(\mathbf{x}, m, \Theta) = \mathbf{r}'\left(\mathbf{x}, \frac{mL}{L'}, \Theta\right),
$$

where $L$ is the trained context length and $L'$ is the length to be extended to. Such simple scaling guarantees equal impact among all tokens, such that each token now takes $L/L'$ of its original position. Since the transformation is merely a change in rotation frequency but not in attention magnitude, it won't harm the original attention pattern, while reducing the speed of decay.

## E    DUALITY BETWEEN RoPE AND MAMBA POSITIONAL ENCODING

We can establish duality between RoPE and Mamba positional encoding mechanisms by showing that both schemes are equivalent to maintaining a bank of keys. They both multiply each channel of the keys by $\exp\left(\sum_{n=j}^{t} \Delta_n\right)$, where $t$ is the current token position (time step in Mamba's context), $j$ is the position associated with a given key, and $\Delta_n$ is a frequency (discretization per step in Mamba's context) term that can vary per-channel and per-position.

**RoPE case:** By construction, the dot product between a given $\mathbf{Q}_t$ and $\mathbf{K}_j$ with RoPE function $r(\cdot)$ is:

$$r(\mathbf{Q}_t, t, \theta) = \mathbf{Q}_t \exp(\mathrm{i}t\theta), \; r(\mathbf{K}_j, j, \theta) = \mathbf{K}_j \exp(\mathrm{i}j\theta),$$

$$\Rightarrow \langle r(\mathbf{Q}_t, t, \theta), r(\mathbf{K}_j, j, \theta) \rangle = \langle \mathbf{Q}_t, \mathbf{K}_j \exp(-\mathrm{i}\theta(t-j)) \rangle$$

$$= \langle \mathbf{Q}_t, \mathbf{K}_j \rangle \cdot \exp\left( \sum_{n=j}^{t} \Delta \right)$$

where (with some slight abuse of terminology) $\theta$ is a vector, and $\exp(\theta)$ is a diagonal matrix containing $\exp(\theta_x)$ for each coordinate $\theta_x$ in $\theta$. Here we keep the computation of RoPE within the complex plane and let $\Delta := -\mathrm{i}\theta$, where $\mathrm{i} = \sqrt{-1}$. In practice we can take the magnitude of the real and imaginary part for the cosine and sine values.

**Mamba case:** The recurrence operation for Mamba can be simplified as: $\mathbf{h}_{t+1} = \exp(-a\Delta_t)\mathbf{h}_t + \Delta_t \mathbf{B}_t \mathbf{x}_t$ (see § B for more details). Unrolling this recurrence gives:

$$\mathbf{h}_t = \sum_{i=1}^{n} \overline{\mathbf{B}}_i \mathbf{x}_i \left( \prod_{k=i+1}^{n-1} \overline{a}_k \right)$$

$$= \sum_{i=1}^{n} \overline{\mathbf{B}}_i \mathbf{x}_i \exp\left( \sum_{k=i+1}^{n-1} -a\Delta_k \right)$$

If we let $\mathbf{C}_t = \mathbf{Q}_t$, $\mathbf{B}_j = \mathbf{K}_j$, $\mathbf{x}_j = \mathbf{v}_j$, and $\Delta'_k = -a\Delta_k$ then the mamba output $\mathbf{C}_t \mathbf{h}_t$ is equivalent to linear attention over $\mathbf{Q}_t, \mathbf{K}_j, \mathbf{v}_j$ with $\mathbf{K}_j$ scaled by $\exp\left( \sum_{k=j+1}^{t} \Delta'_k \right)$, matching our desired formula up to a trivial off-by-one adjustment.

Further circumstantial evidence that RoPE frequencies $\theta$ and Mamba timesteps $\Delta$ are equivalent is the fact that they are, in practice, initialized the same way. RoPE $\theta$ values are spread evenly in log-space between 1 and the inverse of the base frequency, while Mamba's $\mathbf{b}_\Delta$ values are initialized uniformly in log-space between max and min gating thresholds. The failure modes are also the same: slowly-rotating RoPE channel pairs rotate out of distribution when input lengths exceed the training length, while high-frequency channels have completed several rotations and remain in-distribution. Similarly, slowly-converging Mamba heads grow their state magnitude out of distribution, while quickly-converging heads attain their upper bound and then stay stable in perpetuity. These intriguing links may provide avenues for further refinement of Mamba gate-scaling, but we leave this for future work.

## F    EXTENDED DUALITY TABLE AND DISCUSSIONS

The duality between Mamba's decay pattern and RoPE reveals a deeper connection: both Mamba and Transformer encode position by modulating the influence of past inputs, but they do so through different mechanisms. We formalize this connection by extending the Structured State Space Duality (SSD) from Mamba2 (Dao & Gu, 2024) from a position encoding perspective:

> **Structured State Space Duality – A Position Information Encoding Perspective**
> The cumulative decay in Mamba is functionally dual to the explicit position encodings used in Transformers, with both introducing position-dependent biases that progressively de-emphasizing long-range dependencies in language sequence processing.

The detailed extended duality is listed in Table 9. We include new rows on how Transformer's positional embeddings correspond to Mamba's decaying pattern on cumulative forget gates, and identify the corresponding out-of-domain problem that causes bad generalization capability for the model.

**Out-of-Domain (OOD) Problem in Position Encoding.** Mamba's long-context degradation can be interpreted as a form of *out-of-domain (OOD)* failures in its implicit position encoding mechanism. This is similar to observations in RoPE-based transformers, where extrapolating sinusoidal embeddings beyond the training range leads to distorted token interactions and degraded perfor-

Table 9: Extended structured state space duality from a context scaling perspective.

| SSM (Mamba) | Transformer | Interpretation |
|---|---|---|
| $\mathbf{C}$ (contraction matrix) | $\mathbf{Q}$ (queries) | Latent-to-output projection / query |
| $\mathbf{B}$ (expansion matrix) | $\mathbf{K}$ (keys) | Input-to-latent projection / key |
| $\mathbf{X}$ (input sequence) | $\mathbf{V}$ (values) | Projected input tokens / value |
| $\overline{\mathbf{A}}_{s,t} = \begin{cases} 1, & s = t, \\ \prod_{j=s+1}^{t} \bar{a}_j, & s < t, \\ 0, & \text{otherwise.} \end{cases}$ | | Cumulative forget product |
| - lower-triangular shape | $\mathbf{P}$ (causal attention mask) | Causal bias |
| - off-diagonal decay pattern | RoPE or ALiBi | Position-dependent bias |
| $\mathbf{M}_{t,s} = \mathbf{C}_t^{\top} \overline{\mathbf{A}}_{s,t} \overline{\mathbf{B}}_s$ | $\mathbf{T} = \text{Softmax}(\mathbf{Q}\mathbf{K}^{\top} \cdot \mathbf{P})$ | Structured attention matrices |
| $\mathbf{y}_t = \sum_{s=0}^{t} \mathbf{M}_{t,s} \mathbf{X}_s$ | $\mathbf{y}_t = \sum_{s=0}^{t} \mathbf{T}_{t,s} \mathbf{V}_s$ | Weighted sums over the input |
| Per-head SSM kernel | Per-head attention module | Head-specialized feature |
| Head-wise gate scaling (**ours**) | RoPE dimension-wise scaling | Non-uniform domain scaling |
| high-ERF heads | low-frequency dimension | OOD problem |

mance (Shang et al., 2025; Barbero et al., 2025). While RoPE's OOD issue arises from low-frequency

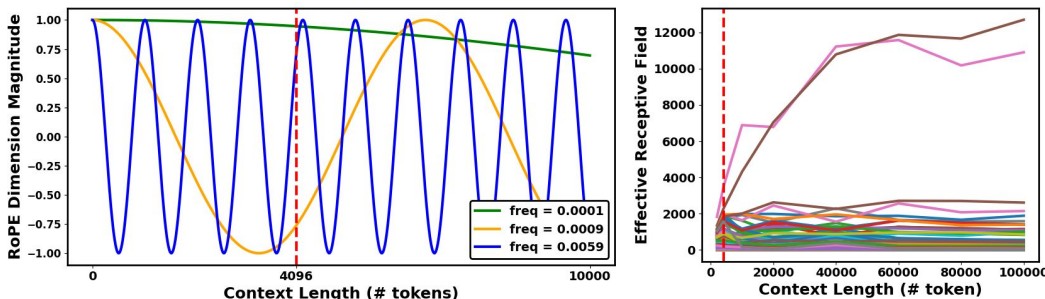

Figure 5: **Left**: RoPE at different frequency. **Right**: Bamba's head-wise ERF change on layer 14, collected from PG-19. The vertical lines show the training context length. For RoPE, low frequency dimensions (dark green) tend to only cover a small portion (1 ∼ 0.95) within trained length. For Mamba, heads with high ERF (pink and brown) that tends to grow much faster than others break the distribution of head magnitude.

rotations in unseen position ranges, Mamba's gated recurrence exhibits a similar failure mode: the cumulative decay function $\overline{\mathbf{A}}_{s,t}$ is under-trained beyond the context window, leading to uncontrolled ERF expansion in some heads: some heads' ERFs plateau, while others grow unchecked; over time, the runaway heads overwhelm the rest. As shown in Fig. 5, both Mamba and RoPE-based Transformers exhibit these pathological behaviors when inputs exceeds their training lengths.

**Connecting Transformer and Mamba CLE methods.** With the extended duality established, we may also give a description on the context extension methods, that was applied to one type of the models, being projected to the other type. Some examples are listed below.

Project Transformer methods to Mamba models:

- NTK-based methods (LocalLLaMA, 2023a;b) & YaRN (Peng et al., 2024): Discriminate heads within different sets of ERF values, and set their input/forget gates with different scaling parameters determined by the input sequence length. Our selective UPI can be categorized as a special case of this type of methods.

- LongRoPE2 (Shang et al., 2025): Perform heuristic search for the optimal combinaiton of the critical ERF to start the scaling, and the scaling parameters for each head with ERF larger than the critical one.

Project Mamba methods to Transformer models:

- Mamba-style Position Embedding (Gu & Dao, 2023; Dao & Gu, 2024): Most current position embedding used by transformers are context agnostic, which is not the case in Mamba. We may build a similar activation-based context-aware position embedding, for more fine-grained position information.
- DeciMamba (Ben-Kish et al., 2025): Decimate a portion of tokens, and only compute a smaller attention map on the rest of the tokens, with the decimated entries in the attention map set to 0 for more concentrated attention.
- LongMamba (Ye et al., 2025): Skip a portion of tokens by copying the closest previous un-skipped token's positional embedding, which effectively keeps all positional embeddings inside the trained domain.

We leave exploring these projected methods as an interesting future work for extending both types of models.

## G   ALTERNATIVE METRICS FOR HEAD SELECTION

Aside from ERF-based top-K% head selection, we also explored alternative metrics including average $\Delta_t$, $\mathbb{E}[\bar{a}_t]$, and the average state growth rate. These metrics are also profiled on the same dataset (Proof-pile ArXiv) used for ERF. $\Delta_t$ can be interpreted as a per-token discretization step controlling

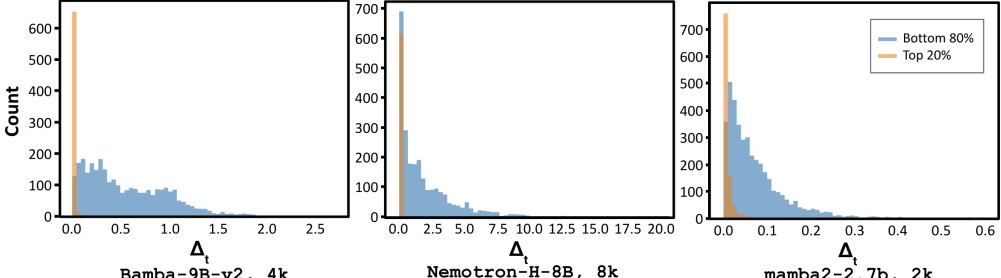

Figure 6: Distribution of average $\Delta_t$ across Mamba heads in different models at their training context length: (**Left**) Bamba-9B-v2 at 4k, (**Middle**) Nemotron-H-8B at 8k, (**Right**) Mamba2-2.7B at 2k. Across all three models, we observe a consistent pattern: heads with high ERF tend to have average $\Delta_t \to 0$, in line with our assumption in § 4.2.

the effective decay rate of the state. Small $\Delta_t$ corresponds to slow state decay and long memory retention, whereas large $\Delta_t$ induces stronger decay and more rapidly changing state dynamics. The distribution of average $\Delta_t$ values for the top-20% ERF heads versus the bottom-80% ERF heads for all three models are shown in Fig. 6. The results show a clear pattern: high-ERF heads exhibit average $\Delta_t$ values tightly concentrated near zero, while low-ERF heads display substantially larger $\Delta_t$ magnitudes. This empirical behavior is consistent with our underlying assumption.

We have also explored using more direct gate-based metrics, such as the average forget gate $\mathbb{E}[\bar{a}_t]$ or the average state growth rate, to identify the unstable heads. Their distributions are shown in Fig. 7 and Fig. 8. In practice, these quantities do not reliably distinguish the problematic heads. For example, many low-ERF heads also exhibit average forget gate values close to 1, yet do not display the large receptive fields or unconverged state behaviors that characterize the truly unstable heads. This suggests that instability is governed by the joint dynamics of both input and output gates, rather than by either gate in isolation. As for the average state magnitude growth, while the majority of the low-ERF heads gather around 1, it does not cleanly separate the two types of heads, and thus requires a similar empirical threshold. We indeed validate that the heads with average large state magnitude growth correspond to large ERF heads, so setting a threshold could result in a similar effect as using ERF.

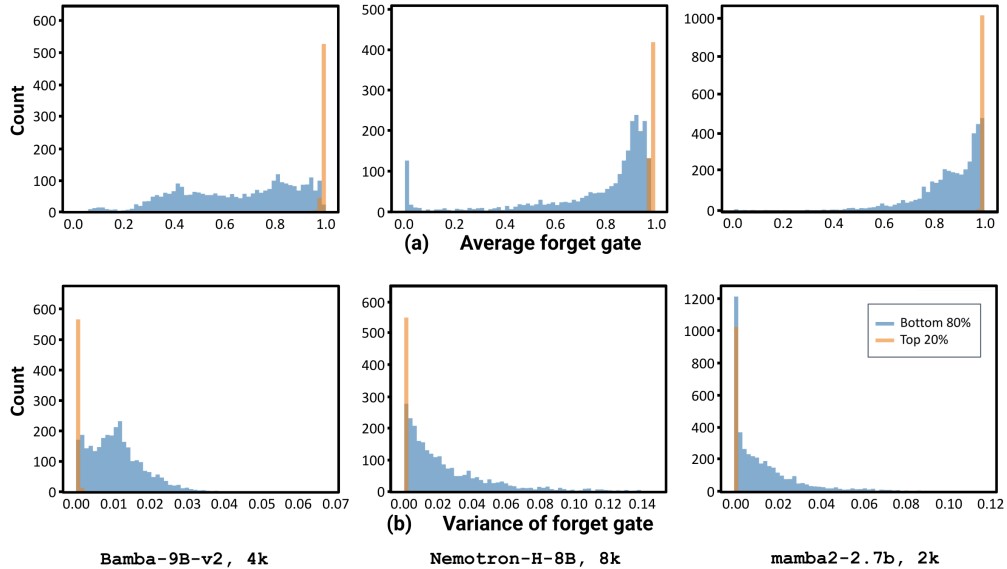

Figure 7: Mean (**a**) and variance (**b**) of forget gates across Mamba heads in different models at their training context length: (**Left**) Bamba-9B-v2 at 4k, (**Middle**) Nemotron-H-8B at 8k, (**Right**) Mamba2-2.7B at 2k. Across all three models, heads with high ERF tend to have average forget gate concentrating at 1 with a close-to-0 variance.

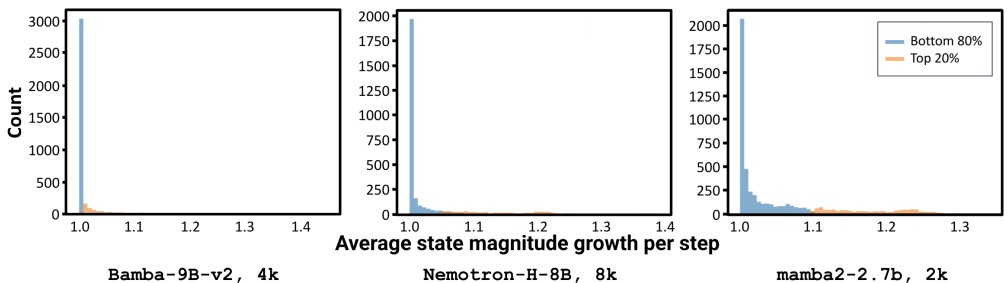

Figure 8: Distribution of average state magnitude growth rate across Mamba heads in different models at their training context length: (**Left**) Bamba-9B-v2 at 4k, (**Middle**) Nemotron-H-8B at 8k, (**Right**) Mamba2-2.7B at 2k. State growth rate needs a similar empirical threshold value to conduct head selection.

The variance of the forget gate can also be an important indicator of state stability. Consider an extreme toy example with two heads: in the first, all gates are 1 except every $n$-th gate is 0; in the second, the gate value is constantly $(n-1)/n$. Although these two heads share the same average gate value, the first has an ERF upper-bounded by $n$, whereas the second can achieve a much larger ERF. The histogram on the variance of forget gates over each head confirms that high ERF heads we identified indeed have smaller variance than the rest.

## H  COMPARISON WITH DECIMAMBA

DeciMamba (Ben-Kish et al., 2025) is, to our knowledge, the first method targeting context extension for Mamba models. While its token-pruning scheme based on SSM step size is quite inspiring, the original DeciMamba pipeline is tailored to Mamba1 (Gu & Dao, 2023), operates over a relatively long training horizon (roughly 1k steps), and is designed in a task-specific manner.

For a fair comparison, we adapted the DeciMamba pipeline to Mamba2-2.7B and fine-tuned the model on the PG-19 training split for 500 steps before evaluating perplexity. As shown in Table 10, UPI is comparable to DeciMamba at shorter contexts for sequence lengths below 16k, but surpasses

| Model | Method | 2k | 4k | 8k | 16k | 32k | 64k |
|-------|--------|------|------|------|-------|-------|-------|
| Mamba2-2.7B | DeciMamba | **6.83** | **7.47** | **8.33** | 15.42 | 34.72 | 67.88 |
| | UPI | 7.40 | 8.21 | 8.75 | **13.69** | **17.58** | **22.24** |

Table 10: Evaluation of Mamba2-2.7B on PG-19 perplexity, comparing the effectiveness of Deci-Mamba and UPI. DeciMamba's token-pruning scheme shows better performance on short context (2k to 8k), while UPI's state growth stabilization improve performance on even longer context.

it at longer contexts, achieving up to 45.64 perplexity points improvement at 64k. We believe this reflects a key difference in design: while DeciMamba achieves strong performance at moderate lengths, its token-pruning scheme does not address the unstable state growth issue that dominates at very long contexts, an issue UPI directly mitigates. We will include this comparison in the final version for completeness.

## I   LLM USAGE

Large language models were used solely for writing assistance. No LLMs were involved in developing the methodology, running experiments, or drawing conclusions.

