# OpenReview forum: "From Collapse to Control: Understanding and Extending Context Length in Emerging Hybrid Models via Universal Position Interpolation"
_ICLR.cc/2026/Conference — ICLR 2026 Poster_

### Official Review · Reviewer_dePa · 2025-10-31

**Soundness:** 3
**Presentation:** 3
**Contribution:** 3
**Rating:** 6
**Confidence:** 3

**Summary:**

This paper addresses context length extension (CLE) for hybrid Mamba-Transformer models, which combine SSM layers with attention layers for efficiency. The authors identify that these models fail beyond training context lengths due to uncontrolled state growth in certain Mamba heads and uneven receptive field contributions. They propose Universal Position Interpolation (UPI), a training-free method that selectively rescales unstable Mamba heads (top 20% by ERF) while applying YaRN to Transformer layers. The method is evaluated on Bamba, Nemotron-H, and Mamba2 models across PG-19, RULER, and LongBench benchmarks.

**Strengths:**

1 - First systematic analysis of context length limitations in hybrid Mamba-Transformer models, addressing a practically relevant issue.

2 - The ERF analysis (Section 3.1) and state magnitude explosion analysis (Section 3.2) provide valuable insights into why hybrid models fail on long contexts.

3 - UPI requires no fine-tuning, making it practically appealing with minimal computational overhead (<3 minutes calibration).

**Weaknesses:**

1.  **Heuristic Head Selection:** The method identifies "unconverged" heads using a proxy: the "top-20% by ERF". While the authors justify this correlation and show its robustness across datasets (Table 6), it remains a heuristic. A more direct, analytical method to identify heads where $E[\overline{a}] \approx 1$ would be more theoretically satisfying.
2.  **The "20%" Threshold:** The 20% threshold is presented as a robust choice, but the ablation (Table 5) is only shown for Bamba-v2. It's unclear if 20% is a "magic number" that generalizes to all models, or if it was re-tuned for Nemotron-H and Mamba2. If it requires per-model tuning, it slightly weakens the "tuning-free" claim (though it's still far cheaper than the baseline).
3.  **Missing Empirical Validation of $\Delta_t$:** The key derivation (Eq 2) relies on the fact that the unstable heads have $\Delta_t \rightarrow 0$. This is a reasonable assumption, but the paper would be strengthened by an empirical validation. A histogram of $\Delta_t$ values for the top-20% ERF heads versus the bottom-80% would provide direct evidence to confirm this crucial assumption.

**Questions:**

1.  The "top-20% ERF" heuristic is practical and effective. Have you explored a more direct method for identifying the unstable heads during calibration? For example, could one directly measure the average forget gate $\overline{a}$ or the state growth rate per head and set a threshold based on that?
2.  How did authors select the 20% threshold for Nemotron-H and Mamba2-2.7B? Was the 20% value from the Bamba-v2 ablation (Table 5) found to be universally optimal, or did this hyperparameter require tuning for each model?
3.  Can authors provide an empirical analysis (e.g., a histogram or density plot) of the $\Delta_t$ distributions for the selected top-20% heads versus the remaining 80%? This would directly validate the core assumption used in the derivation of the simplified input gate scaling (Eq 2).
4.  In Table 3, LongMamba outperforms UPI on the pure Mamba2 model. Authors plausibly argue this is due to LongMamba's task-specific tuning. Could this result also suggest a potential limitation of UPI's "universal" $\Delta_t / n$ scaling when compared to LongMamba's more complex mechanism, especially in a pure-SSM setting?
5.  The RoPE-Mamba duality is a fascinating conceptual bridge. Does this duality suggest that other, more advanced RoPE-scaling techniques (e.g., NTK-aware scaling) could be "translated" to Mamba gate-scaling? What do authors see as the most promising future work based on this "principled bridge"?

---

> ### Author Response · Authors · 2025-11-16
> **Response to Reviewer dePa (1/2)**
>
> Thank you for your positive assessment and insightful suggestions. We respond to each of your comments individually in the following paragraphs:
>
> ---
> > Heuristic Head Selection: The method identifies "unconverged" heads using a proxy: the "top-20% by ERF". While the authors justify this correlation and show its robustness across datasets (Table 6), it remains a heuristic. A more direct, analytical method to identify heads where $E[\overline{a}] \approx 1$ would be more theoretically satisfying.
> >
> > The "top-20% ERF" heuristic is practical and effective. Have you explored a more direct method for identifying the unstable heads during calibration? For example, could one directly measure the average forget gate or the state growth rate per head and set a threshold based on that?
>
> Thank you for the question. We have explored more direct gate-based metrics, such as the average forget gate $\mathbb{E}[\overline{a}_t]$ and the average state growth rate, to identify unstable heads; their distributions are shown in Appendix I.2. In practice, however, **these quantities do not reliably isolate the problematic heads**. For instance, many low-ERF heads also have average forget gate values close to 1, yet they do not exhibit the large receptive fields or unconverged state behaviors of truly unstable heads. This suggests that instability is driven by the **joint dynamics of input and output gates, rather than either gate alone**.
>
> Similarly, while the average state magnitude growth tends to be larger for high-ERF heads, it does not provide a clean separation and still requires choosing an empirical threshold. We confirm that **heads with large average state growth largely overlap with large-ERF heads, so thresholding on state growth would have a similar effect to thresholding on ERF**.
>
> The variance of the forget gate can also be an important indicator of state stability. Consider an extreme toy example with two heads: in the first, all gates are 1 except every $n$-th gate is 0; in the second, the gate value is constantly $(n-1)/n$. Although these two heads share the same average gate value, the first has an ERF upper-bounded by $n$, whereas the second can achieve a much larger ERF. In Figure 8 of Appendix I.2, the histogram on the variance of forget gates over each head confirms that **the forget gate of high ERF heads we identified indeed have much smaller variance than the rest**.
>
> In contrast, ERF captures the **combined effect of these gating dynamics over time**. Empirically, ERF provides a much cleaner separation between stable and unstable heads (as reflected in its bimodal distribution), which is why we adopt it as our identification metric. We will clarify this rationale in the revised version.
>
> ---
> > The "20%" Threshold: The 20% threshold is presented as a robust choice, but the ablation (Table 5) is only shown for Bamba-v2. It's unclear if 20% is a "magic number" that generalizes to all models, or if it was re-tuned for Nemotron-H and Mamba2. If it requires per-model tuning, it slightly weakens the "tuning-free" claim (though it's still far cheaper than the baseline).
> > How did authors select the 20% threshold for Nemotron-H and Mamba2-2.7B? Was the 20% value from the Bamba-v2 ablation (Table 5) found to be universally optimal, or did this hyperparameter require tuning for each model?
>
> Thank you for raising your concern regarding the choice of the top-20%. Please refer to our response to **Common Concern A**.
>
> ---
> > Missing Empirical Validation of $\Delta_t$: The key derivation (Eq 2) relies on the fact that the unstable heads have $\Delta_t\to 0$. This is a reasonable assumption, but the paper would be strengthened by an empirical validation. A histogram of $\Delta_t$ values for the top-20% ERF heads versus the bottom-80% would provide direct evidence to confirm this crucial assumption.
> >
> > Can authors provide an empirical analysis (e.g., a histogram or density plot) of the $\Delta_t$
>  distributions for the selected top-20% heads versus the remaining 80%? This would directly validate the core assumption used in the derivation of the simplified input gate scaling (Eq 2).
>
> Thank you for this insightful suggestion. As recommended, we have added Figure 7 of Appendix I.2,  the histograms of $\Delta_t$ values for the top-20% ERF heads versus the bottom-80% ERF heads for all three models. The results show a clear pattern: **high-ERF heads exhibit $\Delta_t$ values tightly concentrated near zero, while low-ERF heads display substantially larger $\Delta_t$ magnitudes**. This empirical behavior is consistent with our underlying assumption.

---

> ### Author Response · Authors · 2025-11-16
> **Response to Reviewer dePa (2/2)**
>
> > In Table 3, LongMamba outperforms UPI on the pure Mamba2 model. Authors plausibly argue this is due to LongMamba's task-specific tuning. Could this result also suggest a potential limitation of UPI's "universal" $\Delta_t / n$ scaling when compared to LongMamba's more complex mechanism, especially in a pure-SSM setting?
>
> Thank you for pointing out the performance gap on LongBench. Please refer to our response to **Common Concern B**.
>
> ---
> > The RoPE-Mamba duality is a fascinating conceptual bridge. Does this duality suggest that other, more advanced RoPE-scaling techniques (e.g., NTK-aware scaling) could be "translated" to Mamba gate-scaling? What do authors see as the most promising future work based on this "principled bridge"?
>
> Thank you for highlighting the RoPE–Mamba decay duality as a central contribution of our work.
>
> Broadly speaking, yes—**the duality is precisely intended as a principled bridge that allows RoPE-scaling techniques to be “translated” into Mamba gate/decay scaling rules**. In our current work, UPI can be viewed as a relatively *simple, stepwise analogue of YaRN*: our top-20% head-wise linear scaling corresponds to a coarse, head-level version of YaRN’s dimension-adaptive scaling on the RoPE side.
>
> We see several promising directions for future work along this line. First, one could develop more faithful and sophisticated Mamba-side policies that **more closely mirror advanced RoPE-scaling schemes** (e.g., NTK-aware scaling, smoother or multi-stage schedules), rather than our current stepwise approximation. Second, the duality framework naturally **extends to tuning-based approaches**—for example, jointly designing RoPE scaling and Mamba decay scaling under this shared lens, possibly with light additional training to optimize the coupled policy. Finally, we believe the same bridge could **inform the design of richer Mamba variants** (e.g., architectures with more structured or complex-valued state dynamics such as Mamba-3–style layers), where RoPE-inspired scaling principles might guide both the *parameterization of decay* and its *long-context adaptation*.

---

> ### Author Response · Authors · 2025-11-21
>
> Dear Reviewer dePa,
>
> Thank you once again for the time and effort you’ve invested in reviewing our manuscript. We would like to kindly remind you that we have diligently addressed each point raised in your review. We would be more than happy to address any additional concerns or comments you may have.
>
> Thank you!
>
> Best Regards,
>
> Authors of 9337

---

### Official Review · Reviewer_5ThG · 2025-11-01

**Soundness:** 2
**Presentation:** 2
**Contribution:** 2
**Rating:** 4
**Confidence:** 4

**Summary:**

This paper addresses the long-context generalization failure of hybrid Mamba-Transformer models and proposes a training-free context length extension (CLE) method called Universal Position Interpolation (UPI). UPI unifies Mamba’s cumulative decay and Transformer’s rotary positional encoding (RoPE) scaling, enabling hybrid/pure Mamba models to extend context length from 4K/8K to up to 64K tokens without sacrificing short-context accuracy.

**Strengths:**

1. This paper provides the first systematic investigation of long-context failure in hybrid Mamba-Transformer models, addressing a critical gap, prior work focused exclusively on pure Transformer or Mamba architectures, ignoring the unique challenges of hybrid designs. The identification of "uncontrolled state growth" and "uneven receptive field contributions" as root causes is a foundational insight.

2. It balances mathematical rigor (e.g., state magnitude explosion derivation, UPI closed-form proof) and empirical validation. ERF analysis and state growth characterization provide concrete evidence for failure modes, while ablation studies (selective head adjustment, Top-K thresholding) validate the method’s components.


3. Hybrid Mamba-Transformer models are emerging as efficient alternatives to Transformers, but their long-context limitation hinders real-world use (e.g., document understanding, multi-turn dialogue). UPI extends their context window from 4K to 64K tokens without performance loss, making these models viable for long-sequence tasks.

**Weaknesses:**

1. Subjectivity and Non-Generality of the Top-20% High ERF Head Threshold: UPI relies on the rule of "selecting the Top-20% of high ERF heads for adjustment," but this threshold is determined solely based on experimental experience from benchmarks such as PG-19 and RULER, lacking systematic validation. The selection logic for "20%" is not explained, nor is its adaptability to different model architectures (e.g., models with different ratios of Mamba/Transformer layers) and task types (e.g., code generation, multi-turn dialogue) verified.

2. Insufficiently Comprehensive Baselines: Only "LongMamba + YaRN" was selected as the baseline. However, LongMamba is a non-specific baseline designed for pure Mamba and is not compared with "CLE methods specifically optimized for hybrid models" (e.g., Hymba's hybrid head adjustment logic, Samba's state space co-extension). Furthermore, "fine-tuning CLE methods" (e.g., long context fine-tuning for hybrid models) are not compared, making it impossible to quantify the actual value of UPI's "no-training" advantage (e.g., fine-tuning methods may have higher performance; is UPI's deployment advantage sufficient to compensate for the performance gap?).

**Questions:**

1. Rationale for the 20% Top-K Threshold for Mamba Heads.
What is the systematic basis for choosing 20% specifically? Is it derived from theoretical analysis (e.g., the proportion of unstable heads in hybrid models) or just empirical tuning on the tested benchmarks? Does this threshold generalize across different hybrid model architectures (e.g., models with more Transformer layers than Mamba layers) or task types (e.g., code completion vs. multi-document QA)? If the threshold needs adjustment for new models/tasks, what guidelines can be provided?


2. Generalization to Longer Contexts and Unseen Model Configurations
The paper extends context lengths from 2K/4K/8K to 64K, but key gaps remain: Have the authors tested UPI on longer contexts beyond 64K (e.g., 128K or 256K tokens)? If not, does the authors anticipate performance degradation, and what mechanisms might limit scalability?

3. Comparison with More Relevant Baselines for Hybrid Models.
The baseline comparison uses LongMamba (for pure Mamba) + YaRN (for Transformer), but this is a "component-wise combination" rather than a dedicated hybrid CLE method. Have the authors considered baselines that jointly optimize Transformer and Mamba layers for hybrid models?

4. Applicability to Pure Transformer Models and Adaptive Head Selection. The paper frames UPI as "universal," but its utility for pure Transformer models is unaddressed. Can UPI be adapted to pure Transformer models? Since UPI unifies RoPE scaling and Mamba gate dynamics, would applying its core logic (e.g., selective dimension-wise scaling based on ERF-like metrics) outperform existing Transformer CLE methods (e.g., YaRN, LongRoPE)?

---

> ### Author Response · Authors · 2025-11-16
> **Response to Reviewer 5ThG (1/2)**
>
> We sincerely appreciate your encouraging feedback and thoughtful questions. We address each of your points in detail below:
>
> ---
> > Subjectivity and Non-Generality of the Top-20% High ERF Head Threshold: UPI relies on the rule of "selecting the Top-20% of high ERF heads for adjustment," but this threshold is determined solely based on experimental experience from benchmarks such as PG-19 and RULER, lacking systematic validation. The selection logic for "20%" is not explained, nor is its adaptability to different model architectures (e.g., models with different ratios of Mamba/Transformer layers) and task types (e.g., code generation, multi-turn dialogue) verified.
> >
> > Rationale for the 20% Top-K Threshold for Mamba Heads. What is the systematic basis for choosing 20% specifically? Is it derived from theoretical analysis (e.g., the proportion of unstable heads in hybrid models) or just empirical tuning on the tested benchmarks? Does this threshold generalize across different hybrid model architectures (e.g., models with more Transformer layers than Mamba layers) or task types (e.g., code completion vs. multi-document QA)? If the threshold needs adjustment for new models/tasks, what guidelines can be provided?
>
> Thank you for raising your concern regarding the choice of the top-20%. Please refer to our response to Common Concern A .
>
> Beyond the bimodal log-ERF distribution that motivates the 20% cutoff, we also find that this threshold generalizes across architectures and tasks. As shown in Table 8 of Appendix I.3, we evaluate **Nemotron-H-47B**—one of the largest publicly available hybrid SSM–Transformer models—on dialogue and code benchmarks from LongBench v2, as well as a standard passkey retrieval task. Applying the same 20% head-selection pipeline on Nemotron-H-47B yields consistent performance gains, including improvements of **1.7 points on Dialogue History QA** and **6.15 points on Code Repo QA**.
>
> ---
> > Only "LongMamba + YaRN" was selected as the baseline. However, LongMamba is a non-specific baseline designed for pure Mamba and is not compared with "CLE methods specifically optimized for hybrid models" (e.g., Hymba's hybrid head adjustment logic, Samba's state space co-extension).
> >
> > Comparison with More Relevant Baselines for Hybrid Models. The baseline comparison uses LongMamba (for pure Mamba) + YaRN (for Transformer), but this is a "component-wise combination" rather than a dedicated hybrid CLE method. Have the authors considered baselines that jointly optimize Transformer and Mamba layers for hybrid models?
>
> Thank you very much for these constructive comments. We clarify that UPI is, to the best of our knowledge, **the first interpolation method designed to extend the context length of existing hybrid Mamba–Transformer models** via the RoPE–Mamba decay duality (Section 4.3). A key property of UPI is that it operates purely at inference time and requires no architectural changes or additional training, which makes it directly applicable to off-the-shelf hybrid models.
>
> In contrast, Hymba and Samba propose **new hybrid SSM–Transformer architectures** (in the spirit of Bamba and Nemotron-H): Hymba *mixes SSM and attention at the head level*, and Samba replaces full attention with *sliding-window attention*. **Their long-context behavior results from training these modified architectures, rather than from a generic, post-hoc CLE mechanism**. Because their context length is primarily determined by the training context window and architectural choices, they are not directly comparable as CLE baselines in our setting, which targets **inference-time extension of already-trained hybrid models**.
>
> Given the absence of prior CLE methods for hybrid architectures, we chose LongMamba + YaRN as a strong, model-agnostic baseline that can be applied directly to hybrid models without retraining. We will revise the paper to clarify this distinction and to position UPI as complementary to hybrid architectures such as Hymba and Samba, which could incorporate UPI as an inference-time CLE component.

---

> ### Author Response · Authors · 2025-11-16
> **Response to Reviewer 5ThG (2/2)**
>
> > Furthermore, "fine-tuning CLE methods" (e.g., long context fine-tuning for hybrid models) are not compared, making it impossible to quantify the actual value of UPI's "no-training" advantage (e.g., fine-tuning methods may have higher performance; is UPI's deployment advantage sufficient to compensate for the performance gap?).
>
> Thank you for raising this important point. The goal of UPI is not to replace long-context fine-tuning, but to **provide a post-hoc, training-free CLE mechanism that can be applied to existing hybrid Mamba–Transformer models with only modest profiling data and no retraining**. UPI is therefore *conceptually orthogonal* to fine-tuning-based approaches and can, in principle, be applied with them.
>
> To illustrate complementarity, we fine-tuned Bamba-9B-v2 to 32k and 128k context lengths. As shown in Table 10 of Appendix I.3, applying UPI to the 32k model consistently improves RULER scores across all lengths, with noticeable gains of 5.4 at 64k and 10.4 at 128k, partially closing the gap to the 128k fine-tuned checkpoint. Moreover, even the 128k model benefits from UPI: beyond 16k, UPI yields additional improvements of roughly 1–3 while only slightly affecting very short contexts. Together, these results show that **UPI remains useful even when strong long-context fine-tuning is available, providing complementary gains on top of fine-tuned extrapolation**.
>
> ---
> > Generalization to Longer Contexts and Unseen Model Configurations The paper extends context lengths from 2K/4K/8K to 64K, but key gaps remain: Have the authors tested UPI on longer contexts beyond 64K (e.g., 128K or 256K tokens)? If not, does the authors anticipate performance degradation, and what mechanisms might limit scalability?
>
> Thank you for this question. As also observed in prior interpolation-based CLE methods (e.g., position interpolation, YaRN, LongRoPE, LongMamba), **tuning-free gains diminish once the target context exceeds roughly 4× the training or fine-tuning context length**. Several works have noted that **addressing positional out-of-distribution alone is necessary but not sufficient condition for strong long-context performance**: beyond a certain point, the model becomes fundamentally constrained by its pre-training distribution and capacity.
>
> We therefore expect tuning-free CLE methods, including UPI, to encounter *inherent limitations* far beyond the pre-training window. At extreme extrapolation ranges, the model encounters positional patterns that are far outside its training distribution, which leads to very low performance for both the base and extended model. Consequently, it is hard to draw meaningful conclusions in the 128k–256k range.
>
> While extending hybrid models to 128K-1M contexts is an important long-term goal, it remains a largely open challenge for all tuning-free CLE approaches. As the first work to study CLE for hybrid SSM-Transformer models, our focus in this paper is on **establishing the core failure analysis and principled training-free extension methods**. Scaling beyond this regime is a natural direction for future work.
>
> ---
> > Applicability to Pure Transformer Models and Adaptive Head Selection. The paper frames UPI as "universal," but its utility for pure Transformer models is unaddressed. Can UPI be adapted to pure Transformer models? Since UPI unifies RoPE scaling and Mamba gate dynamics, would applying its core logic (e.g., selective dimension-wise scaling based on ERF-like metrics) outperform existing Transformer CLE methods (e.g., YaRN, LongRoPE)?
>
> Thank you for raising this point. We would like to clarify that UPI is not intended as a new, standalone CLE policy for pure Transformer models. Rather, **UPI provides a duality framework connecting RoPE-based position interpolation in Transformers with decay dynamics in Mamba layers**.
>
> Concretely, UPI derives its Mamba-side adjustments by translating the dimension-wise scaling used in Transformer CLE methods (e.g., YaRN, LongRoPE) through the RoPE-Mamba decay duality. On the Transformer side, UPI is fully compatible with existing methods CLE techniques: in a pure Transformer model, one would simply apply a strong RoPE-scaling method such as YaRN and LongRoPE directly. On the Mamba side, the same duality motivates a head-wise decay adjustment, where our **ERF-based head selection serves as the SSM-side analogue of dimension-adaptive RoPE scaling**.
>
> Therefore, the  “universality” of UPI refers to its ability to **apply a single conceptual principle consistently across both Mamba and Transformer layers in hybrid architectures**, not to claim that it outperforms Transformer-specific CLE methods when applied to pure Transformer models. Designing new Transformer-only CLE policies is outside our scope, and existing Transformer methods remain more appropriate in that setting.

---

> ### Author Response · Authors · 2025-11-21
>
> Dear Reviewer 5ThG,
>
> Thank you once again for the time and effort you’ve invested in reviewing our manuscript. We would like to kindly remind you that we have diligently addressed each point raised in your review. We would be more than happy to address any additional concerns or comments you may have.
>
> Thank you!
>
> Best Regards,
>
> Authors of 9337

---

> ### Comment · Reviewer_5ThG · 2025-11-22
>
> Dear Authors of 9337,
>
> Thanks for the author's reply. I think the author's reply is convincing, and I will slightly adjust the score.
>
> Best,
> Reviewer 5ThG

---

> > ### Author Response · Authors · 2025-11-22
> >
> > Thank you for your careful re-evaluation after the rebuttal. We appreciate your engagement and are glad that our clarifications resolved your concerns.

---

### Official Review · Reviewer_iUod · 2025-11-01

**Soundness:** 3
**Presentation:** 4
**Contribution:** 4
**Rating:** 6
**Confidence:** 4

**Summary:**

- This paper addresses the Context Length Extension (CLE) problem for hybrid Mamba-Transformer models. While these hybrid models are computationally efficient and perform well within their training context window, the authors show they "collapse" catastrophically when presented with sequences longer than what they were trained on (e.g., perplexity spikes when a 4K-trained model sees 8K tokens).
- The failure is not just due to the Transformer components, but primarily stems from a subset of "unstable" or "unconverged" heads within the Mamba layers. These heads have forget gates consistently close to 1, causing their internal state to grow uncontrollably beyond the training length. This "state magnitude explosion" overwhelms the outputs of other, stable heads, leading to a "feature collapse" where most of the model's neurons become inactive.
- The authors introduce UPI, a training-free, closed-form method to scale the context window of hybrid models. UPI works by:
  - Using a small calibration dataset (100 samples, <3 minutes) to find the top 20% of Mamba heads with the largest Effective Receptive Field (ERF), which are the unstable ones.
  - For these selected heads, it scales their discretization step size $\Delta_t$ by a factor of $\frac{1}{n}$ (where $n$ is the context scaling factor). This effectively "slows down" the state updates, preventing explosive growth.
  - It seamlessly integrates with existing Transformer CLE methods like YaRN to handle the positional encodings in the Transformer layers.
- UPI is evaluated on models like Bamba and Nemotron-H, extending their effective context from 4K/8K up to 64K tokens. It achieves substantial reductions in perplexity and improves performance on long-context benchmarks (RULER, LongBench) without any fine-tuning, outperforming baselines that combine methods like LongMamba and YaRN.

**Strengths:**

- The paper is well written and easy to follow.
- The paper provides the first systematic analysis of why hybrid models fail on long contexts.
- UPI is a simple, lightweight, and highly effective. Its training-free nature and minimal computational overhead for calibration.
- The experiments are comprehensive, using multiple benchmarks (PG-19, LongBench, RULER) and models. The results are convincing, showing dramatic improvements in perplexity and task accuracy at extended contexts.

**Weaknesses:**

- The choice of the "top-20% of heads by ERF" is shown to be effective but is ultimately a heuristic. While the paper shows this selection is robust across datasets, a more theoretical or analytical justification for this specific threshold would strengthen the method.
- The largest model tested is 9B parameters. While this is a reasonable scale, validating the method on larger hybrid models would be important to confirm its scalability.
- The results on the pure Mamba2 model show that UPI is outperformed by the LongMamba baseline on the LongBench benchmark.
- The tables do not include comparison to DeciMamba [1].

[1] DeciMamba: Exploring the Length Extrapolation Potential of Mamba

**Questions:**

- How well does this method scale to larger models?
- Could you please include a comparison with DeciMamba?

---

> ### Author Response · Authors · 2025-11-16
> **Response to Reviewer iUod**
>
> We really appreciate your positive feedback and constructive suggestions/questions. We have addressed each of your comments in detail below:
>
> ---
> >The choice of the "top-20% of heads by ERF" is shown to be effective but is ultimately a heuristic. While the paper shows this selection is robust across datasets, a more theoretical or analytical justification for this specific threshold would strengthen the method.
>
> Thank you for raising your concern regarding the choice of the top-20%. Please refer to our response to **Common Concern A** .
>
> ---
> > The largest model tested is 9B parameters. While this is a reasonable scale, validating the method on larger hybrid models would be important to confirm its scalability.
> >
> > How well does this method scale to larger models?
>
> Thank you for the suggestion. **We would like to clarify that 9B-scale models are already among the largest publicly available SSM-Transformer architectures. Prior CLE work for Mamba has primarily focused on much smaller models (e.g., 1.4B for LongMamba, 2.8B for DeciMamba).** To further validate scalability, we applied UPI to one of the largest hybrid model currently released, *Nemotron-H-47B-Base-8k*, and evaluated it on a standard passkey retrieval task at multiple context lengths (8k, 16k, 32k) and the Dialogue History QA task and the Code Repo QA task from LongBench v2. As shown in Table 8 of Appendix I.3, UPI **improves accuracy by 6.15% on the coding task, 1.7% higher on the dialogue task**, and recovers substantial performance on long-context passkey retrieval over the 47B model, which demonstrates that UPI scales effectively to substantially larger hybrid architectures. We will incorporate these results into the final version to provide a more comprehensive evaluation.
>
> ---
> > The results on the pure Mamba2 model show that UPI is outperformed by the LongMamba baseline on the LongBench benchmark.
>
> Thank you for pointing out the performance gap on LongBench. Please refer to our response to **Common Concern B**.
>
> ---
> > The tables do not include comparison to DeciMamba [1].
> >
> > Could you please include a comparison with DeciMamba?
>
> Thank you for highlighting this related work. DeciMamba is, to our knowledge, the first method targeting context extension for Mamba models. While its token-pruning scheme based on SSM step size is quite inspiring, the original DeciMamba pipeline is tailored to Mamba1, operates over a relatively long training horizon (~1k steps), and is designed in a task-specific manner.
>
> For a fair comparison, we adapted the DeciMamba pipeline to Mamba2-2.7B and fine-tuned the model on the PG-19 training split for 500 steps before evaluating perplexity. As shown in Table 9 of Appendix I.3, UPI is comparable to DeciMamba at shorter contexts for sequence lengths below 16k, but surpasses it at longer contexts, achieving up to **45.64 perplexity points improvement at 64k**. We believe this reflects a key difference in design: while DeciMamba achieves strong performance at moderate lengths, *its token-pruning scheme does not address the unstable state growth issue that dominates at very long contexts*, an issue UPI directly mitigates. We will include this comparison in the final version for completeness.

---

> ### Author Response · Authors · 2025-11-21
>
> Dear Reviewer iUod,
>
> Thank you once again for the time and effort you’ve invested in reviewing our manuscript. We would like to kindly remind you that we have diligently addressed each point raised in your review. We would be more than happy to address any additional concerns or comments you may have.
>
> Thank you!
>
> Best Regards,
>
> Authors of 9337

---

> > ### Comment · Reviewer_iUod · 2025-11-21
> >
> > The authors have satisfactorily addressed all of my concerns in the revised manuscript. Therefore, I have updated my final score accordingly.

---

> > > ### Author Response · Authors · 2025-11-21
> > >
> > > Thanks for your detailed review and for revisiting the paper in light of our responses. Your feedback helped us improve the revision, and we’re grateful for your updated assessment.

---

### Author Response · Authors · 2025-11-16
**Response to All Reviewers**

We thank all reviewers for their thoughtful and constructive feedback. We are encouraged that reviewers highlighted the **originality** of our systematic analysis of hybrid Mamba–Transformer long-context failures and the UPI framework (iUod, 5ThG, dePa), the **practical impact** of enabling long-context use (up to 64k) without fine-tuning (iUod, 5ThG, dePa), and the **simplicity and ease of use** of UPI as a lightweight, training-free method with minimal calibration (iUod, dePa). We first address common questions raised by multiple reviewers, followed by responses to specific reviewer comments.

---
## Common Concern A (iUod, 5ThG, dePa) – Rationale behind ERF-based top-K% head selection

The first common question raised by all reviewers concerns our head-wise selective scaling mechanism, where we select the top-K% of Mamba heads across the model for scaling. Empirically, we find that setting K = 20% works well as a consistent choice across multiple models (Mamba2-2.7B, Bamba-9B-v2, Nemotron-H-8B) and benchmarks (PG-19 perplexity, RULER, LongBench).

While the paper did not elaborate on this choice, the 20% cutoff directly reflects the empirical distribution of log-ERF scores across heads. In all evaluated models, **the log-ERF distribution is bimodal**: a minority subset of high-ERF heads (approximately 10~30%, depending on layer and sequence length) separates cleanly from the majority of low-ERF heads. Exact percentages vary by model, but we adopt 20% as a general and effective cutoff because it closely aligns with the average split point of the bimodal log-ERF distribution and is further supported by the empirical results in Table 5. To support this explanation, we have added the corresponding log-ERF distributional analysis for all three models to Appendix I.1 of the updated submission, and we will integrate this discussion to Section 4.4 of the main paper.

We also note that the cutoff can, in principle, be determined directly from the log-ERF distribution itself. Because the log-ERF scores consistently exhibit a bimodal structure, one can identify the boundary between the two modes (for example, via Gaussian mixture modeling or other clustering-based criteria) and select the high-ERF cluster accordingly. In practice, this data-driven boundary aligns closely with the 20% cutoff we adopt, which is why we choose the simpler heuristic. That said, developing a fully automated, model- or task-adaptive cutoff selection method is an interesting direction for future work.

---
## Common Concern B (iUod, dePa) – Performance Difference vs. LongMamba + YaRN Baseline on LongBench

We note that LongMamba achieves its best LongBench results only after extensive, *benchmark-specific* hyperparameter search (approximately 10 hours per 7B-scale model in our setup), using direct evaluation on LongBench during tuning. In contrast, UPI requires only about 100 forward passes (around 3 minutes in the same setup) to configure a new model and produces strong LongBench performance despite having no access to the benchmark. Given this difference in tuning cost and oracle access, a small gap on LongBench is expected.

Importantly, on RULER, which is another long-context benchmark, Mamba2 + UPI outperforms Mamba2 + LongMamba by 4.9% at 4K and 2.9% at 8K sequence length, which demonstrates that UPI has an advantage: it avoids expensive, task-specific tuning while still matching or surpassing LongMamba on certain evaluations.

Task-specific designs offer another promising direction for extending UPI. For example, one could replace our generic profiling dataset with a task- or domain-specific corpus and then apply a similar “ERF profiling plus head-selective scaling” procedure. However, since the primary goal of this work is to analyze failure modes and introduce the duality framework, we leave such task-tailored extensions to future work.

---
### Appendix Updates

We append updated tables and figures to the end of the appendix: I.1 presents the bimodal log-ERF distribution that motivates our top-K% head selection; I.2 provides analysis of alternative head-selection indicators, as suggested by Reviewer dePa; and I.3 reports additional evaluations on (1) a larger model (Nemotron-H-47B) and diverse tasks (dialogue and code), (2) comparison with DeciMamba, and (3) complementarity with fine-tuning–based extrapolation. **We have highlighted all changes from the original submission in the latest revision.**

---

> ### Author Response · Authors · 2025-11-17
> **Changes Highlighted in the Latest Revision**
>
> We have updated the manuscript and highlighted all changes relative to the original submission in the latest revision. We thank the reviewers again for their constructive feedback and are happy to clarify any remaining questions.

---

### Author Response · Authors · 2025-12-01
**Summary for AC after Score Rollback (1/2)**

Dear Area Chair,

We are writing regarding our ICLR submission 9337 about the recent score rollback and AC reassignment due to the leakage incident. We fully understand and respect the decision, and we wanted to briefly summarize the paper, prior reviews, and what we added in the rebuttal so that you have a concise view of the current status.

---

## 1. One-sentence summary of the paper
We provide the **first systematic analysis of long-context failures in hybrid Mamba-Transformer models** and introduces **UPI, a training-free, closed-form context length extension method** that extends hybrid models from 4K/8K up to 64K contexts without fine-tuning, by selectively rescaling unstable Mamba heads through a **RoPE-Mamba decay duality**.

---

## 2. Review trajectory before the global rollback

Initial overall scores: 6, 4, 6

After the rebuttal and discussion period:
- Reviewer iUod explicitly stated that all concerns were satisfactorily addressed and raised their score from 6 to 8.
- Reviewer 5ThG found our rebuttal convincing and raised their score from 4 to 6.
- Reviewer dePa already had a positive assessment (6) and did not raise further questions.

Therefore, before the global reset, the review scores were **6, 4, 6 => 8, 6, 6**, with two reviewers explicitly acknowledging that their concerns were resolved based on the revised manuscript and rebuttal.

---

## 3. What we did in the rebuttal and revision
We addressed four main concerns in the rebuttal:
1. *the seemingly heuristic ERF-based top-20% head selection*, by showing a clear bimodal log-ERF split across models and explaining 20% as a simple proxy for a data-driven cutoff;
2. *the performance gap to LongMamba + YaRN and missing DeciMamba comparison*, by clarifying LongMamba’s benchmark-specific tuning, adding RULER results where UPI outperforms LongMamba, and directly comparing to DeciMamba on PG-19;
3. *scalability and generality of the 20% rule*, by demonstrating consistent gains on the 47B Nemotron-H across multiple long-context tasks;
4. *the relationship to long-context fine-tuning*, by showing that UPI still yields meaningful improvements when applied on top of 32K/128K fine-tuned Bamba-9B-v2, indicating that UPI is complementary rather than a replacement.

The details are listed below.

---

### Rationale behind ERF-based top-20% head selection (iUod, 5ThG, dePa).
**Concern: The “top-20% by ERF” head selection looked heuristic and under-explained.**

Our response and changes:
- We justified our choice of scaling the top 20% of Mamba heads by showing that, across all evaluated models, **the log-ERF scores form a bimodal distribution** where roughly 20% high-ERF heads clearly separate from the bulk.
- We added log-ERF distribution plots for all three models to Appendix I.1 and clarified that 20% closely matches this empirical split and yields robust gains across PG-19, RULER, and LongBench (Table 5).
- We discussed how an automated cutoff can in principle be derived directly by clustering the log-ERF distribution (e.g., Gaussian mixture modeling), and framed 20% as a simple, robust approximation that aligns with this data-driven boundary.

---

### Performance difference vs. LongMamba + YaRN on LongBench (iUod, dePa).
**Concern: (i) LongMamba + YaRN vs. UPI, especially on LongBench for pure Mamba2. (ii) Missing comparison to DeciMamba.**

Our response and changes:
- We clarified that LongMamba’s best LongBench results rely on *extensive, benchmark-specific hyperparameter search with direct LongBench access*, whereas UPI is configured in **a few minutes from a small profiling set without using the benchmark**, explaining the small performance gap.
- We also added RULER results showing that Mamba2 + UPI outperforms Mamba2 + LongMamba by 4.9% at 4K and 2.9% at 8K, highlighting that UPI can match or surpass LongMamba while avoiding expensive, task-specific tuning.
- We adapted DeciMamba to Mamba2-2.7B and added a direct comparison (up to 45.64 points improvement on PG-19 perplexity), which highlights that UPI directly tackles the unstable state growth issue that dominates at long contexts.


---

### Scalability to larger models and additional tasks (iUod, 5ThG).
**Concern: Does UPI scale beyond 9B models? Does the 20% threshold generalize to larger architectures and different tasks?**

Our response and changes:
- We evaluated **Nemotron-H-47B**, one of the largest publicly available hybrid SSM-Transformer models, on passkey retrieval, dialogue history QA, and code repo QA.
- Using exactly the same 20% head-selection method, UPI improves accuracy by 1.7 points on Dialogue History QA and 6.15 points on Code Repo QA, and recovers performance on long-context passkey retrieval (Appendix I.3, Table 8)
- These results demonstrate that UPI and the 20% rule scale to substantially larger hybrid models and diverse task types.

---

> ### Author Response · Authors · 2025-12-01
> **Summary for AC after Score Rollback (2/2)**
>
> ### Complementary with long-context fine-tuning (5ThG)
> **Concern: Without comparing to long-context fine-tuning, the value of “training-free” CLE is hard to quantify.**
>
> Our response and changes:
> - We fine-tuned **Bamba-9B-v2 to 32K and 128K contexts** and then applied UPI on top of these fine-tuned checkpoints.
> - UPI improves RULER scores even on the 32K fine-tuned model, with gains of 5.4 at 64K and 10.4 at 128K, partially closing the gap to the 128K checkpoint. It also yields 1-3 point gains on top of the 128K fine-tuned model (Appendix I.3, Table 10).
> - This shows that UPI is complementary to long-context fine-tuning rather than a competing replacement.
>
> ---
>
> ## 4. Summary
> After considering these clarifications and new results, **both iUod and 5ThG explicitly stated that their concerns were resolved and raised their scores accordingly**. We recognize that the leakage incident and rollout have placed additional burden on ACs and the program chairs, and we appreciate the efforts going into ensuring a fair process. Our aim with this summary is simply to give you a concise view of the paper and the post-rebuttal changes, and to highlight that two reviewers already revisited and upgraded their assessments based on the current revision.
>
> We would of course be happy to answer any further questions or provide additional clarifications if that would be helpful for your evaluation.
>
> ---
>
> Thank you very much for your time and consideration.
>
> Best regards,
>
> Authors of submission 9337

---

### Meta-Review · Area_Chair_Szbp · 2026-01-12

**Summary:**

The paper does fundamental analysis of length generalization issues of SSMs and proposes an effective solution that shows good results in practice. All reviewers appreciate the contribution. The initial concerns were around the 20% heads heuristic and limited comparisons. Authors response effectively addresses these concerns. Hence I suggest acceptance.

**Reviewer Concerns:**

Reviewer concerns were around the 20% heads heuristic and limited comparisons.

**Reviewer Scores:**

5ThG 4 -> 6

---

### Decision · Program_Chairs · 2026-01-26

Accept (Poster)